# Characterization and functional analysis of cathelicidin-MH, a novel frog-derived peptide with anti-septicemic properties

Jinwei Chai[1,2†], Xin Chen[1†], Tiaofei Ye[1,2], Baishuang Zeng[2], Qingye Zeng[2], Jiena Wu[2], Barbora Kascakova[3], Larissa Almeida Martins[4], Tatyana Prudnikova[3], Ivana Kuta Smatanova[3], Michail Kotsyfakis[4], Xueqing Xu[2]*

[1]Department of Respiratory Medicine, Zhujiang Hospital, Southern Medical University, Guangzhou, China; [2]Guangdong Provincial Key Laboratory of New Drug Screening, School of Pharmaceutical Sciences, Southern Medical University, Guangzhou, China; [3]Faculty of Science, University of South Bohemia in Ceske Budejovice, Branisovska, Czech Republic; [4]Institute of Parasitology, Biology Center of the Czech Academy of Sciences, Branisovska, Czech Republic

*For correspondence:
xu2003@smu.edu.cn

†These authors contributed equally to this work

Competing interests: The authors declare that no competing interests exist.

**Abstract** Antimicrobial peptides form part of the innate immune response and play a vital role in host defense against pathogens. Here we report a new antimicrobial peptide belonging to the cathelicidin family, cathelicidin-MH (cath-MH), from the skin of *Microhyla heymonsivogt* frog. Cath-MH has a single α-helical structure in membrane-mimetic environments and is antimicrobial against fungi and bacteria, especially Gram-negative bacteria. In contrast to other cathelicidins, cath-MH suppresses coagulation by affecting the enzymatic activities of tissue plasminogen activator, plasmin, β-tryptase, elastase, thrombin, and chymase. Cath-MH protects against lipopolysaccharide (LPS)- and cecal ligation and puncture-induced sepsis, effectively ameliorating multiorgan pathology and inflammatory cytokine through its antimicrobial, LPS-neutralizing, coagulation suppressing effects as well as suppression of MAPK signaling. Taken together, these data suggest that cath-MH is an attractive candidate therapeutic agent for the treatment of septic shock.

## Introduction

Septic shock represents an unbalanced host response to infection and remains a major cause of death in intensive care units in developed countries (*Huang et al., 2019*). Severe sepsis is associated with tissue damage, abnormal coagulation and immune responses, and 'cytokine storms' triggered by inflammatory factors (*Huang et al., 2019*). Lipopolysaccharide (LPS; endotoxin) is the main component of the outer membrane of Gram-negative bacteria which, as one of the most potent natural immunostimulatory compounds, plays a crucial role in sepsis and septic shock through hyperactivation of the innate immune system (*Salomao et al., 2012*). LPS-stimulated endotoxemia is mediated by Toll-like receptor-4 (TLR-4), which causes overwhelming production of tissue-damaging cytokines tumor necrosis factor-α (TNF-α), interleukin-1 (IL-1), and interleukin-6 (IL-6) and free radicals (*Salomao et al., 2012*; *Van Amersfoort et al., 2003*). Despite decades of research, there are still no effective preventative therapies for sepsis-induced death, and with the rising incidence and mortality of sepsis (*Huang et al., 2019*), new drugs to decrease the morbidity and mortality of septicemia are urgently required.

Antimicrobial peptides (AMPs) are a key first line of defense against pathogen invasion (*Varga et al., 2018*). Cathelicidin is a family of AMPs present in monocytes, lymphocytes, natural killer cells, and epithelial cells in various mammalian species. Cathelicidins are derived from precursor proteins consisting of an N-terminal signaling peptide and a highly conserved cathelin domain in

front of a C-terminal mature peptide (*Agier et al., 2015*). In addition to their antimicrobial, anti-inflammatory, and immunomodulatory properties, cathelicidins, particularly those adopting helical structures in lipids, possess diverse biological activities including LPS neutralization, antioxidant, direct chemotaxis, and wound healing effects (*Cao et al., 2018*; *Mu et al., 2017*; *Scott et al., 2002*; *Wei et al., 2013*; *Wu et al., 2018*). For example, the well-characterized human cathelicidin LL-37 inhibits neutrophil chemotaxis in sepsis-induced acute lung injury, macrophage activation and cytokine production, and endothelial cell apoptosis in Gram-negative infections and improves the survival of polybacterial septic mice (*Agier et al., 2015*; *Dürr et al., 2006*; *Hu et al., 2016*; *Qin et al., 2019*). Thus, cathelicidin represents an important therapeutic candidate for sepsis after traditional therapy (*Huang et al., 2019*). Cathelicidin-AL is the first amphibian cathelicidin identified in *Amolops loloensis* in 2012 (*Hao et al., 2012*) and, since then, over 20 cathelicidin AMPs have been discovered in 11 amphibian species (*Cao et al., 2018*; *Hao et al., 2012*; *Ling et al., 2014*; *Mu et al., 2017*; *Qi et al., 2019*; *Wei et al., 2013*; *Wu et al., 2018*; *Yang et al., 2017*; *Yu et al., 2013*). *Microhyla heymonsivogt* frogs mainly inhabit Southern China and survive in the wild by arming themselves with multiple biomolecules against various pathogens (*Varga et al., 2018*), but no *M. heymonsivogt* AMP has yet been identified.

Since some amphibian cathelicidins have shown antimicrobial and anti-inflammatory properties, we investigated whether *M. heymonsivogt* harbors a cathelicidin. We identified the novel cathelicidin cathelicidin-MH (cath-MH) from the skin of *M. heymonsivogt* frogs. Cath-MH contains one α-helix structure in membrane-mimetic environments and has favorable antimicrobial, anti-inflammatory, LPS-binding, and protease interference activities. In addition, cath-MH markedly decreased the mortality and inhibited pathological abnormalities and inflammatory cytokine expression in tissues of LPS- and cecal ligation and puncture (CLP)-induced septic mice. Therefore, Cath-MH represents a novel potential anti-inflammatory therapy, particularly for bacterial sepsis.

## Results

### Identification and characterization of cath-MH

Based on PCR-based cDNA cloning, a novel AMP was obtained from the cDNA library derived from the skin of *M. heymonsivogt* frogs. As illustrated in *Figure 1A*, the 693 bp cDNA contained a complete open reading frame and the deduced amino acid precursor consisted of 172 amino acid residues including a predicted 22-residue signal peptide, a representative 105-residue cathelin domain, and a 39-residue mature peptide. NCBI Basic Local Alignment Search Tool (BLAST) analysis and multisequence alignment revealed that the cath-MH precursor had high sequence similarity with other members of the cathelicidin family (*Figure 1B*), containing four cysteine residues at the end of the cathelin domain and a highly variable C-terminal mature region. According to the amino acid sequences of known frog cathelicidins, the predicted mature cath-MH sequence (APCKLGCKIKKVK QKIKQKLKAKVNAVKTVIGKISEHLG) possessed 12 strongly basic residues distributed across the entire molecule, resulting in an 11 charge at pH 7.0. The molecular weight was 4.26 kDa, and isoelectric point was 10.32. Sequence alignment with other amphibian cathelicidins revealed that mature cath-MH also contained a typical intramolecular pentapeptide ring-forming disulfide bond (*Figure 1C*). In addition, sequence alignment also demonstrated 22.5% sequence identity and 50% sequence similarity between cath-MH and human LL-37 (PDB ID: 5NMN) (*Sancho-Vaello et al., 2017*; *Figure 1D*).

### Phylogenetic analysis

A phylogenetic tree was constructed with 14 amphibian cathelicidin precursors from 12 amphibian species using the neighbor-joining method (*Figure 1—figure supplement 1*). Their sequences formed three distinct groups: the bottom cluster comprised three sequences identified in species belonging to Salamandridae and Ranidae; the middle cluster contained three sequences from two species belonging to Bufonidae; while the third cluster was comparatively complex and consisted of 12 peptides identified in different genera. Cath-MH was classified into the bottom cluster according to its phylogenetic relationships and was not closely related to other frog cathelicidins, indicating that it may have some different functions.

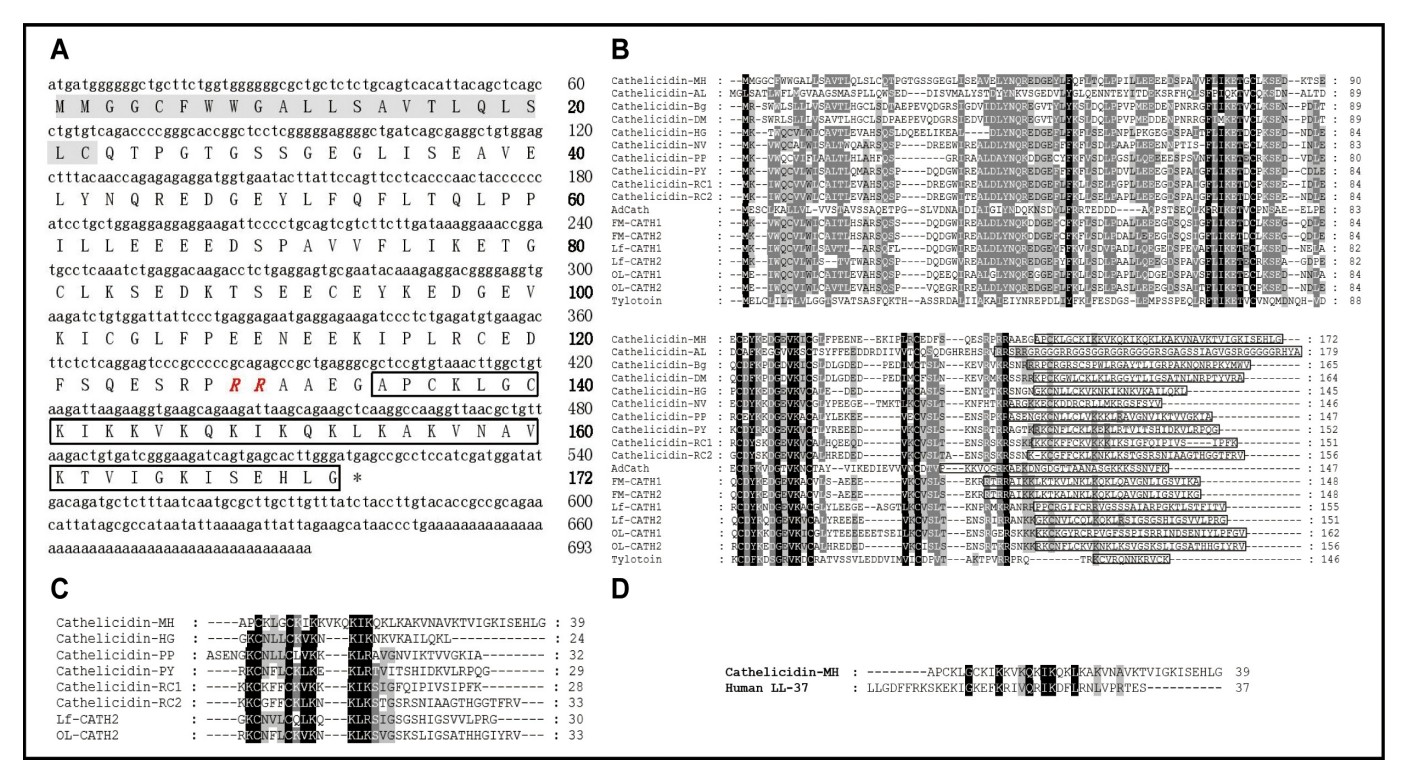

**Figure 1.** Sequence characterization of cath-MH. (**A**) The cDNA and deduced amino acid sequence of cath-MH. The signal peptide is emphasized in gray, and the RR residues in red italic indicate the end of an acidic spacer domain. The sequence of mature peptide is boxed, and the stop codon is denoted by an asterisk (*). Nucleotide and amino acid numbers are displayed after the sequences. Multisequence alignment of cathelicidins from (**B**) Amphibians or (**C**) frogs. In order to maximize structural similarity, residue deletions denoted by the hyphens (-) have been introduced in some sequences and the identical residues are indicated in black. The highly conserved residues are shaded. The domains of putative mature peptides are boxed. (**D**) Sequence alignment of cath-MH and LL-37.

The online version of this article includes the following figure supplement(s) for figure 1:

**Figure supplement 1.** Evolutionary relationships of cath-MH and other amphibian cathelicidins.

## Secondary structure of cath-MH

The optimal concentration of cath-MH for crystallization was 2.719 mg/ml. The small trigonal-shaped crystals of the peptide were successfully grown under Crystal ScreenHT (Hampton Research) conditions of 2 M ammonium sulfate over 60 d at 277 K (*Figure 2—figure supplement 2*). While cath-MH crystals diffracted X-rays to 1.95 Å resolution, the crystal structure was solved at 2.20 Å resolution because the initial resolution did not provide a complete non-protein $2F_o - F_c$ electron density map. The peptide crystals belonged to the rhombohedral space group *R*32. The crystal asymmetric unit contained one molecule of the peptide with a solvent content of 41.76%. As shown in (*Figure 2A*) (PDB ID: 7AL0), the final cath-MH structure consisted of all 39 amino acid residues forming one α-helix composed of 31 amino acids (from L5 to S32) with two short loops marked as loop-1 and loop-2. Loop-1 was formed by the first four amino acids A1-K4 at the N-terminal part of the structure, and loop-2 contained the last four amino acids E36-G39 at the C-terminal part (*Figure 2A*). Non-protein electron density was interpreted as 13 water molecules and two chloride ions. The stabilization of the α-helical monomer was provided by a disulfide bridge formed from sulfur atoms of C3 and C7 and by several H-bonds between O atom C3 and Nζ atom K8 at 3.15 Å; between O atom Q25 and Oγ1 T29 at 2.70 Å; between O atom of I31 and Oγ S35 at 2.92 Å; and between Nζ K33 and Oε1 E36 at 2.68 Å distance. Further stabilization was realized by interactions between Sγ atom C3 with Nζ atom of K11; O atom of Q14 and Oε1 Q18; and Nζ atom of K17 and Nζ atom of K21 (*Figure 2B*). The monomer was more positively charged at the N-terminus than the C-terminus (*Figure 2C*) and shared high secondary structural similarity (*Figure 2D*). In addition, the X-ray

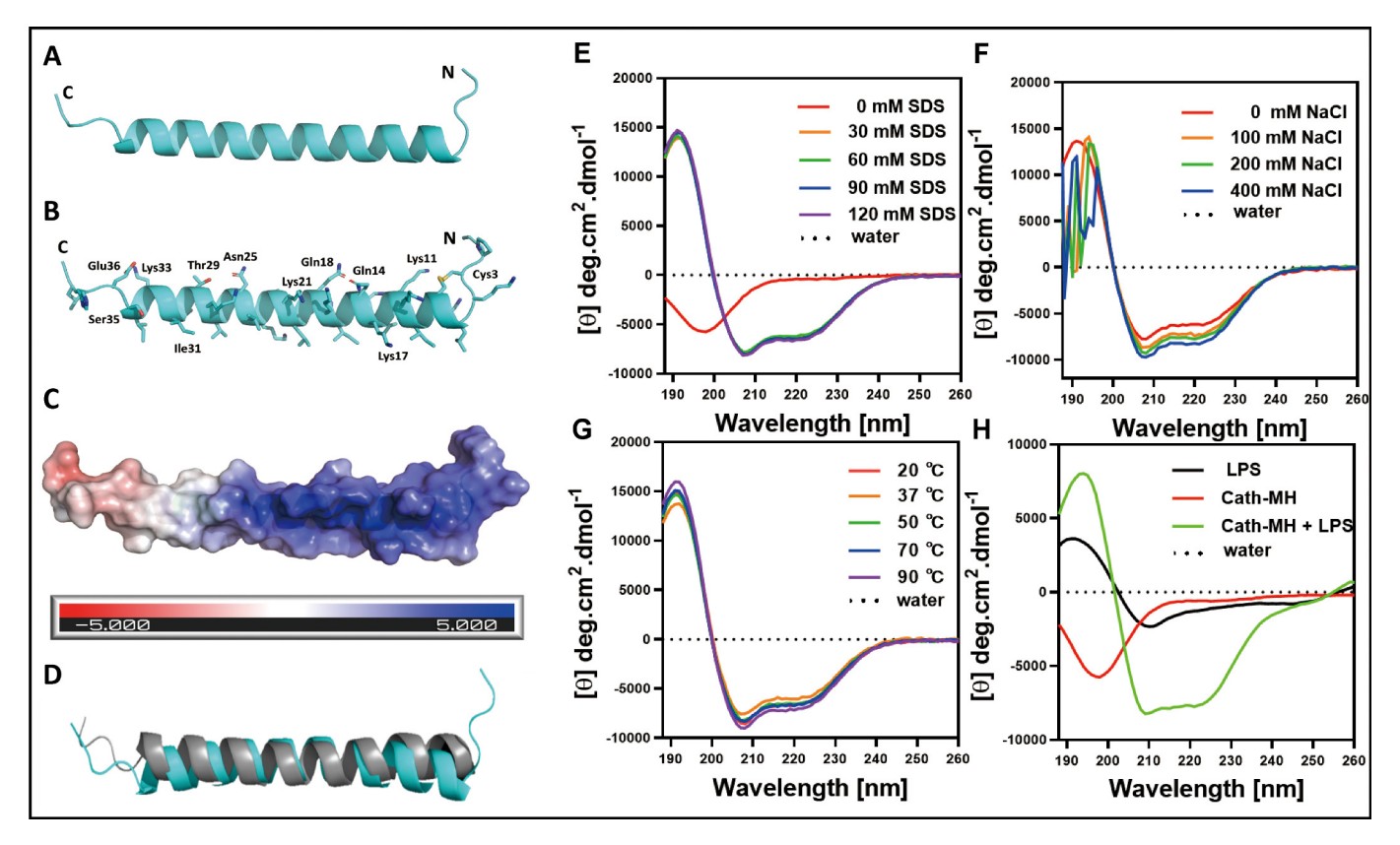

**Figure 2.** The secondary structure of cath-MH. (**A**) Cartoon representation of the cath-MH structure with labeled N and C termini. (**B**) Cartoon visualization of residues participating in the stabilization of the helix. Amino acid side chains are shown as sticks. (**C**) Electrostatic potential of the monomeric structure of cath-MH mapped on the molecular surface. The negative surface potential is colored red while the positive one is colored blue. (**D**) Cartoon representation of superposition of the cath-MH (shown in cyan) and LL-37 (PDB ID: 5NMN; shown in gray) structures. CD spectra of 0.213 mg/ml cath-MH dissolved in the indicated concentrations of SDS solutions (**E**) or in 60 mM SDS plus the indicated concentrations of NaCl (**F**) or in 60 mM SDS at 20, 50, 70, and 90℃ for 1 hr (**G**) or, in 5 mM HEPES plus 50 µM LPS (**H**) (n = 3 per group).

The online version of this article includes the following figure supplement(s) for figure 2:

**Figure supplement 1.** The synthesis and characterization of cath-MH.

**Figure supplement 2.** The cath-MH crystals grown from 2 M ammonium sulfate used for diffraction analysis.

**Figure supplement 3.** Visualization of the cath-MH dimeric structure.

structure revealed that the N-terminus was ordered in an α-helix and that a short loop formed from the first four residues due to disulfide bridge formation between the sulfur atoms of C3 and C7 and additional H-bond interactions provided by K11, Q14, Q18, K17, and K21 amino acid side chains.

## Circular dichroism analysis

Circular dichroism (CD) analysis was performed with the synthesized peptide to elucidate the secondary structure of cath-MH and determine its stability in different solutions. As illustrated in *Figure 2E* and *Supplementary file 1*, $H_2O$-dissolved cath-MH presented negative peaks at 198 nm in CD spectra, suggesting the presence of a random coil structure. A large positive peak at 191 nm accompanied by two small negative peaks at 208 nm and 224 nm were observed in the CD spectra of cath-MH dissolved in SDS solutions, suggesting that the secondary structure of cath-MH contained an α-helix. Moreover, different concentrations of SDS (30–120 mM) slightly changed the secondary structure components of cath-MH. Dissolution in SDS solutions with NaCl varied the cath-MH CD spectra (*Figure 2F*), although the α-helix structure of cath-MH was preserved in CD spectra of 400 nM NaCl solution. The secondary structure of the peptide remained highly similar at different

temperatures (*Figure 2G*). Interestingly, compared to the CD spectra seen with LPS solution or peptide dissolved in $H_2O$, the characteristic α-helix peaks of cath-MH in the presence of 50 µM LPS were consistent with those seen in SDS solutions (*Figure 2H*), suggesting that LPS solutions could induce an α-helical secondary structure of cath-MH, which might be associated with its antimicrobial activity.

## Antimicrobial activity of cath-MH

The antimicrobial properties of cath-MH were investigated against multiple microorganisms. Cath-MH had broad-spectrum antimicrobial activity against all tested organisms (*Figure 3—figure supplement 1A*), with *Propionibacterium acnes* ATCC 6919 (MIC 6.65 µg/ml) the most sensitive. To further explore the antimicrobial efficiency of cath-MH, its bacterial killing kinetics were determined using a colony counting assay. As shown in *Figure 3—figure supplement 1B*, at concentrations of 1× and 2× MICs, cath-MH was unable to completely eliminate all the microbes in 180 min, but 3× MIC of peptide successfully eliminated all viable colonies by 150 min, suggesting that cath-MH is bactericidal rather than bacteriostatic. As a positive control, meropenem took about 120 min to completely clear *Escherichia coli* ATCC 25922 at 1× MIC (*Figure 3—figure supplement 1B*).

The antimicrobial action of cath-MH was also tested at different NaCl concentrations and temperatures. As shown in *Figure 3—figure supplement 1C*, compared with the control without NaCl, twofold increase in MIC of cath-MH against *E. coli* ATCC 25922 after 1 hr of incubation with 50–400 mM NaCl at room temperature and at increasing temperatures from 37 to 90℃ compared to control at 20℃. Notably, cath-MH MICs increased twofold to eightfold in a concentration-dependent manner when incubated at 37℃ with human serum.

## Effect of cath-MH on bacterial cell membranes

The binding interaction between cath-MH and microorganisms was first explored by flow cytometry. As presented in *Figure 3A*, FITC-labeled cath-MH successfully bound to *S. aureus* ATCC 25923 and *E. coli* ATCC 25922 as demonstrated by the difference in fluorescence intensity after 30 min incubation compared to the control group without treatment. It is well known that AMPs exert their activity by permeating the cytoplasmic membrane and disrupting membrane integrity, thereby releasing intracellular inclusions and causing cell death. Confocal laser scanning microscopy and scanning electron microscopy were performed to determine whether cath-MH exerted such an effect and to explore its antimicrobial mechanisms. As shown in *Figure 3B*, compared to the control group without peptide treatment, dead bacteria staining red with propidium iodine (PI) increased in both strains after 90 min treatment with 1× MIC cath-MH, indicating that cath-MH might destroy cell membrane integrity, leading to permeation of PI into microbes. In agreement with the confocal laser scanning microscopy, scanning electron microscopy revealed that while microbes retained their normal shape and smooth surface in the control group, microorganisms treated with cath-MH (1× MIC) for 30 min were obviously swollen with cell surface deformations (*Figure 3C*). These results together indicated that, like many AMPs, cath-MH permeabilized the cells and disrupted membrane integrity.

## Interaction between cath-MH and LPS

The agglutination activity of cath-MH was tested with *E. coli* ATCC 25922. As shown in *Figure 4A*, agglutination occurred in the presence of cath-MH after 1 hr of incubation compared to controls. However, agglutination was suppressed by LPS and D-(+)-galacturonic acid; agglutination activity of the peptide was therefore caused by its direct binding to LPS on Gram-negative bacteria. Consistent with this, the concentration-dependent bactericidal activity of cath-MH against *Bacillus subtilis* CMCC 63501 was suppressed with increasing LPS concentrations and was even completely abrogated by >150 µM LPS, with lg colony forming unit (CFU) values of bacteria incubated with cath-MH at the three tested concentrations comparable to those without peptide (*Figure 4—figure supplement 1*). Thus, cath-MH can bind to LPS to suppress its antimicrobial activity against *B. subtilis* CMCC 63501 in a concentration-dependent manner.

Isothermal titration calorimetry (ITC) was next performed to determine the binding reaction between cath-MH and LPS. As illustrated in (*Figure 4B*), binding of cath-MH to LPS decreased the enthalpy and ITC profiles, suggesting the release of heat during the reaction. The $K_d$ used to determine the binding affinity between cath-MH and LPS was 5.53 µM, indicating strong affinity.

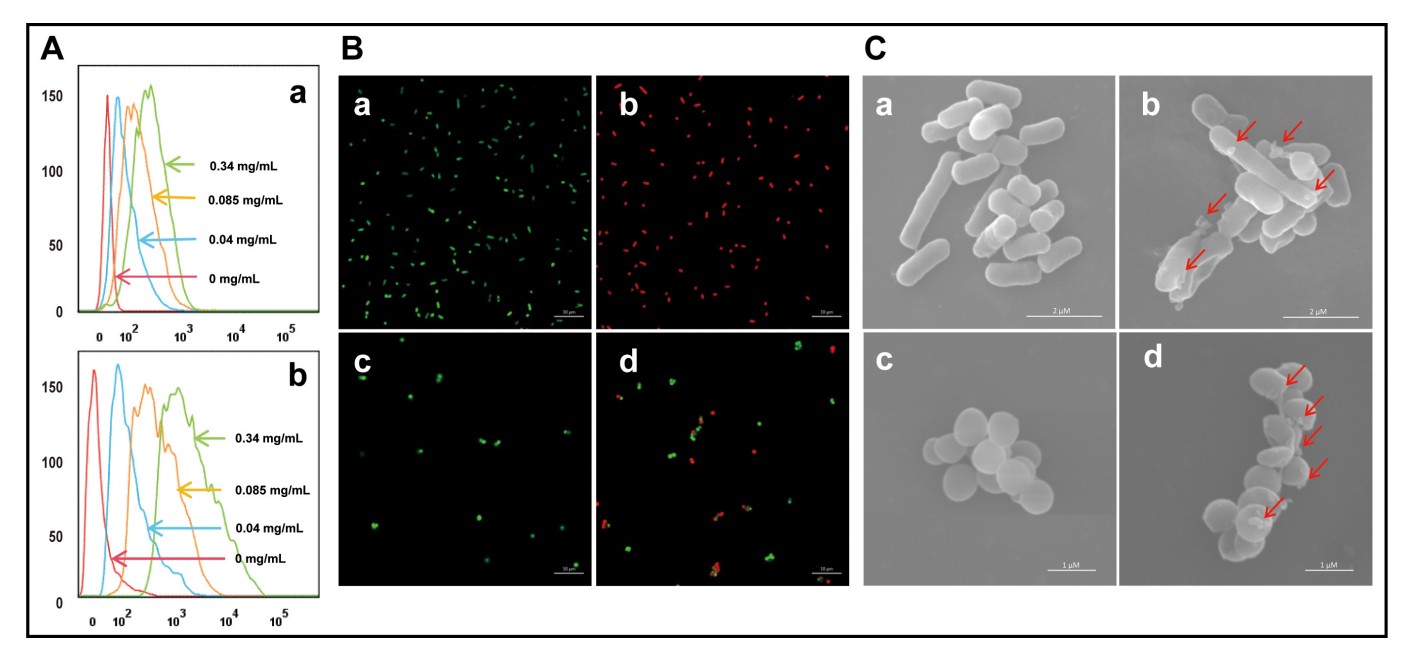

**Figure 3.** Effect of cath-MH on bacterial cell membranes. (**A**) Flow cytometry analysis of interaction between FITC-labeled cath-MH and *E. coli* ATCC 25922 (**a**) or *S. aureus* ATCC 25923 (**b**). (**B**) Confocal laser scanning microscopy of control *E. coli* ATCC 25922 (**a**), *S. aureus* ATCC 25923 (**c**), and cath-MH-treated *E. coli* ATCC 25922 (**b**), *S. aureus* ATCC 25923 (**d**). Cath-MH is at concentration of 1× MIC. Live and dead cells stained by SYTO9 and PI are presented as green and red fluorescence, respectively. Scale bars: 10 µm. (**C**) Scanning electron microscopy of control *E. coli* ATCC 25922 (**a**), *S. aureus* ATCC 25923 (**c**), and cath-MH-treated *E. coli* ATCC 25922 (**b**), *S. aureus* ATCC 25923 (**d**). Red arrows indicate membrane defects. Scale bars: 2 µm in *E. coli* ATCC 25922; 1 µm in *S. aureus* ATCC 25923 (n = 3 per group).

The online version of this article includes the following figure supplement(s) for figure 3:

**Figure supplement 1.** Antimicrobial activity of cath-MH.

Furthermore, the change in enthalpy (ΔH) of binding between cath-MH and LPS was −1.76 ± 0.93 kcal/mol. As a potential component of LPS, the reaction of D-(+)-galacturonic acid with cath-MH was also investigated. As shown in (*Figure 4C*), there was also heat release when the peptide bound to D-(+)-galacturonic acid (ΔH −3.01 ± 0.047 kcal/mol; $K_d$ 1.29 ± 0.144 µM). Binding saturated at about 10 min with a molar ratio of 6 (peptide to D-(+)-galacturonic acid) (*Figure 4C*). These results suggest that D-(+)-galacturonic acid and LPS are both reactive with the cath-MH. The binding reaction of cath-MH was further investigated by surface plasmon resonance imaging (SPRi). As illustrated in (*Figure 4D,E*), there was a strong SPRi signal and an increase in resonance units in a concentration-dependent manner, suggesting direct binding of cath-MH to both LPS and D-(+)-galacturonic acid. The $K_d$ values of cath-MH binding to LPS and D-(+)-galacturonic were about 2.81 µM and 4.52 µM, respectively, in agreement with the ITC results (*Supplementary file 4*; *Figure 4D and E*).

## Effects of cath-MH on plasma recalcification time and enzymatic activity

Cath-MH dose-dependently inhibited plasma coagulation (*Figure 5A*). Cath-MH increased the $Ca^{2+}$ clotting time of normal human plasma in vitro, and even cath-MH at 0.4 µg/ml delayed clotting to ~3 min (*Figure 5A*). As shown in *Figure 5B*, an intense SPRi signal was produced and the resonance unit increase was concentration dependent when three plasma concentrations were passed over chip-immobilized cath-MH, indicating that some plasma components can bind cath-MH. Screening assays were performed to examine the inhibitory activity of cath-MH against a panel of 13 serine proteases. As shown in *Figure 5C*, at 0.004 mg/ml, cath-MH showed little or no activity against trypsin, FXIIa, FXIa, FXa, and kallikrein but significantly inhibited the enzymatic activities of tPA, plasmin, β-tryptase, elastase, and thrombin by 50.7%, 17.7%, 82.0%, 11.1%, and 8.6%, respectively. Interestingly, cath-MH increased the enzymatic activity of chymase, and its remaining enzymatic activity in the presence of cath-MH was 341.5 ± 7.7%. In agreement with these findings, when these six

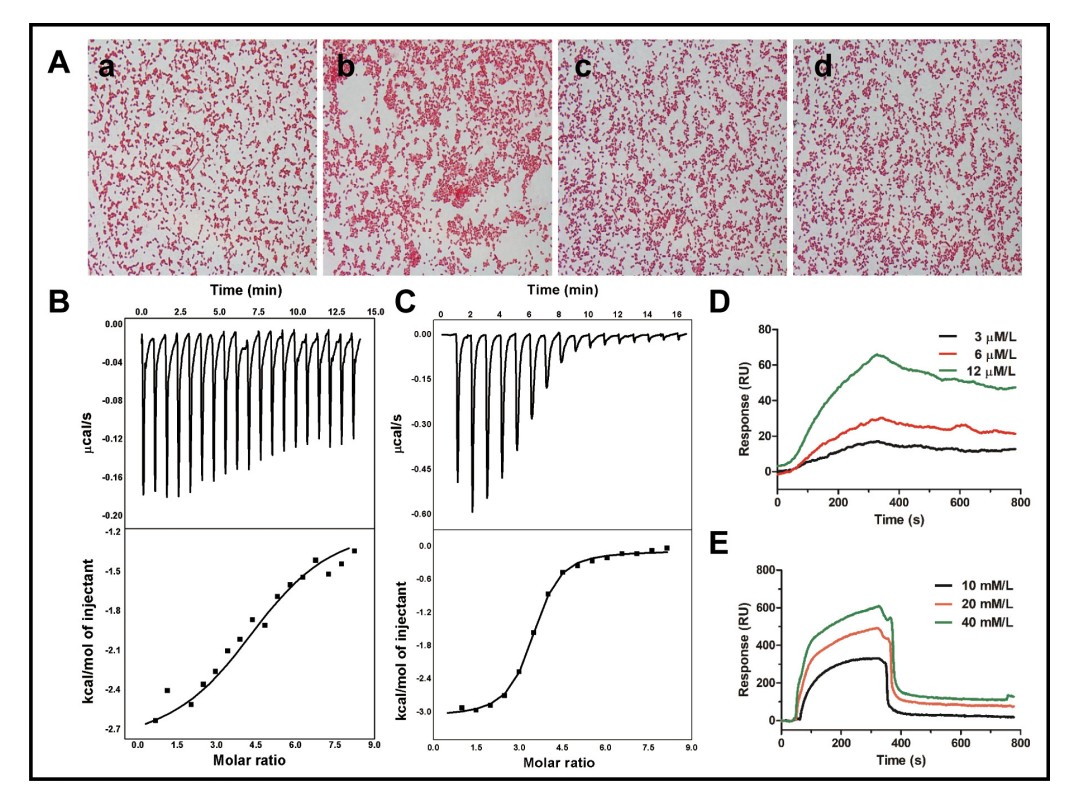

**Figure 4.** Binding reaction of cath-MH with LPS and D-(+)-galacturonic acid. (A) Agglutination of *E. coli* ATCC 25922 treated with cath-MH. *E. coli* ATCC 25922 at a density of $1 \times 10^9$ CFU/ml were treated with BSA (a) and 1× MIC of cath-MH (b); 1× MIC of cath-MH plus an equal volume of 1 mg/ml LPS (c); or 4 mM D-(+)-galacturonic acid (d) for 30 min at room temperature before being Gram stained (migration: 1000×). ITC analysis of cath-MH binding to (B) LPS and (C) D-(+)-galacturonic acid. The top panels show thermo changes of each injection at different time points, while the bottom panels show changes in enthalpy as a function of ligand/target molar ratio. Data were fitted using a single-site binding model using MicroCal Origin software. SPRi analysis of (D) LPS and (E) D-(+)-galacturonic acid binding to cath-MH immobilized on a gold chip (n = 2 per group).

The online version of this article includes the following figure supplement(s) for figure 4:

**Figure supplement 1.** The effect of LPS on the antimicrobial activity of cath-MH against *B. subtilis* CMCC 63501.

proteases were applied to sensor chip-immobilized cath-MH, there were concentration-dependent increases in resonance units, confirming the interaction with specific enzymes and indicating a high affinity interaction between cath-MH and tPA, plasmin, β-tryptase, elastase, thrombin, and chymase (*Figure 5—figure supplement 1*). In consistent with in vitro results, the injection of 5 mg/kg cath-MH delayed the bleeding time of mice for 5.34 min when compared the saline water control group (*Figure 5D*).

## Effects of cath-MH in LPS-induced septic mice

Mice injected with LPS were observed for 72 hr, and the typical signs of sepsis such as diarrhea, pyuria, and death were monitored. These signs obviously decreased in mice treated with cath-MH (5 mg/kg) or dexamethasone (10 mg/kg). Furthermore, the mortality rate was 90% in the model group by the end of experiment, while dexamethasone and cath-MH treatments successfully reduced LPS-induced mortality to 80% and 70%, respectively (*Figure 6A*). The effect of cath-MH on pathological changes, MPO activity, and pro-inflammatory cytokine expression was also evaluated. Histopathological examination of the organs revealed a protective effect for cath-MH in septic mice (*Figure 6B*, *Figure 6—figure supplement 1*). After 6 hr of LPS injection, mice exposed to LPS alone exhibited significant inflammatory injuries in multiple organs such as hemorrhage, severe edema, inflammatory cell infiltration, and tissue destruction in lungs; hemorrhage, hepatocellular necrosis, inflammatory cell infiltration in livers; and tubular renal cell swelling and hemorrhage in kidneys. However,

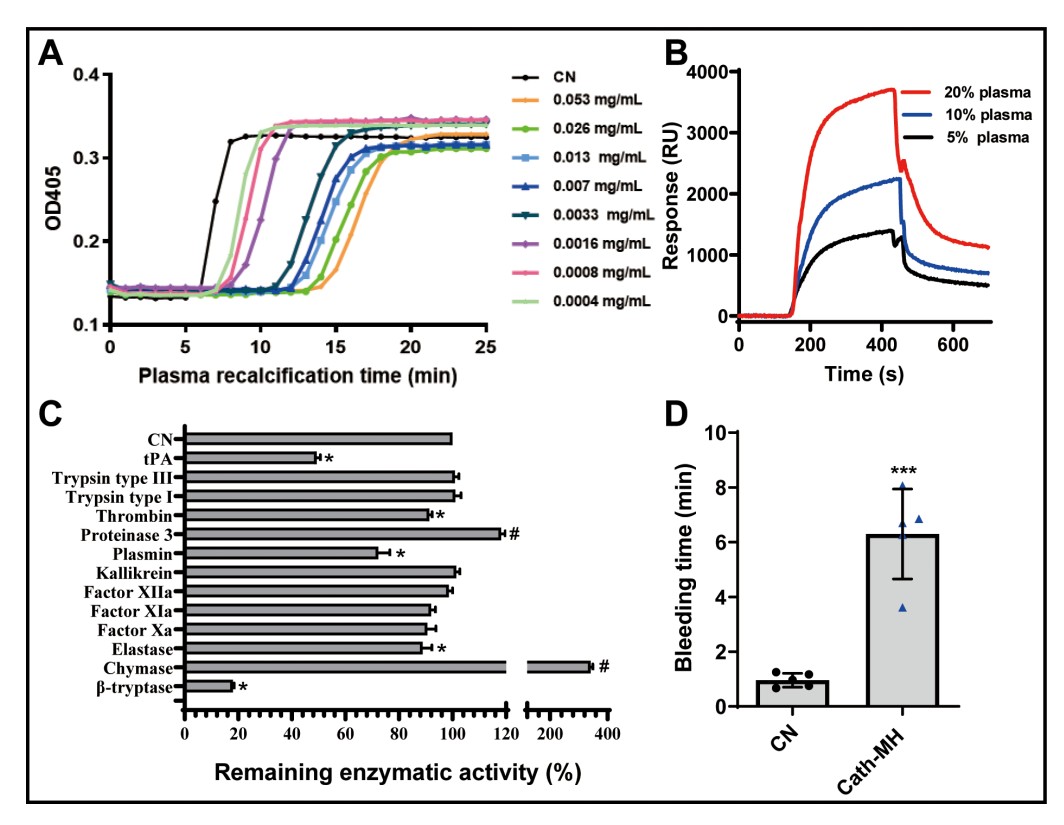

**Figure 5.** Effects of cath-MH on plasma recalcification time (PRT) and enzymatic activity. (A) Effect of cath-MH on normal human PRT. The indicated concentrations of cath-MH in 50 μl of 0.15 M NaCl were incubated with 50 μl of normal human platelet-free plasma and 50 μl of 25 mM CaCl₂. Data are reported as mean ± SEM (n = 3 per group). (B) Representative SPRi sensograms of human plasma binding to cath-MH immobilized on a gold chip. (C) Inhibitory activity cath-MH against specific proteases. 0.004 mg/ml cath-MH was examined against 13 different proteases in triplicate. Enzyme concentrations are described in **Supplementary file 5**. Bars indicate the mean remaining enzymatic activity in the presence of cath-MH. #p<0.05 and *p<0.05, significantly different compared to control group without cath-MH. (D) Effect of cath-MH on bleeding time in vivo (2 mm segment of the tail). CN: control. Data are reported as mean ± SEM (n = 3 per group). ***p<0.001, significantly different compared to control group without cath-MH. The online version of this article includes the following figure supplement(s) for figure 5:

**Figure supplement 1.** Representative SPRi sensograms of cath-MH immobilized in a gold chip binding to the indicated different proteases.

treatment with cath-MH or dexamethasone 1 hr after LPS challenge markedly ameliorated those abnormal histopathological changes and significantly decreased the injury scores in the lungs, livers, and kidneys compared to LPS administration alone (p<0.05) (*Figure 6B,C*, *Figure 6—figure supplement 1*). In agreement, the wet to dry lung weight ratio in the model group increased dramatically in contrast to the control group, while cath-MH and dexamethasone obviously suppressed this increase compared to the model group (*Figure 6D*). MPO activity also increased markedly in the model group compared to the control group and was significantly inhibited by cath-MH and dexamethasone (by 60.6 ± 5.5% and 94.5 ± 4.2%, respectively; *Figure 6E*). As shown in *Figure 6F–K*, *Figure 6—figure supplements 2* and *3*, increases in gene expression and protein production of pro-inflammatory cytokines in different organs elicited by LPS after 12 hr were significantly inhibited by cath-MH and dexamethasone, except for TNF-α protein production in livers.

To further explore the protective mechanism of cath-MH in LPS-induced endotoxic shock, the expression of ERK, JNK, and p38 in lungs, livers, and kidneys were investigated by western blotting (*Figure 7*). Consistent with their effects on pro-inflammatory cytokine expression in lungs, livers, and kidneys, injection of LPS (25 mg/kg) significantly upregulated the expression of phosphorylated ERK, JNK, and p38 in all tested tissues. However, treatment with cath-MH and dexamethasone successfully suppressed these increases but had no influence on total ERK, JNK, and p38 expression.

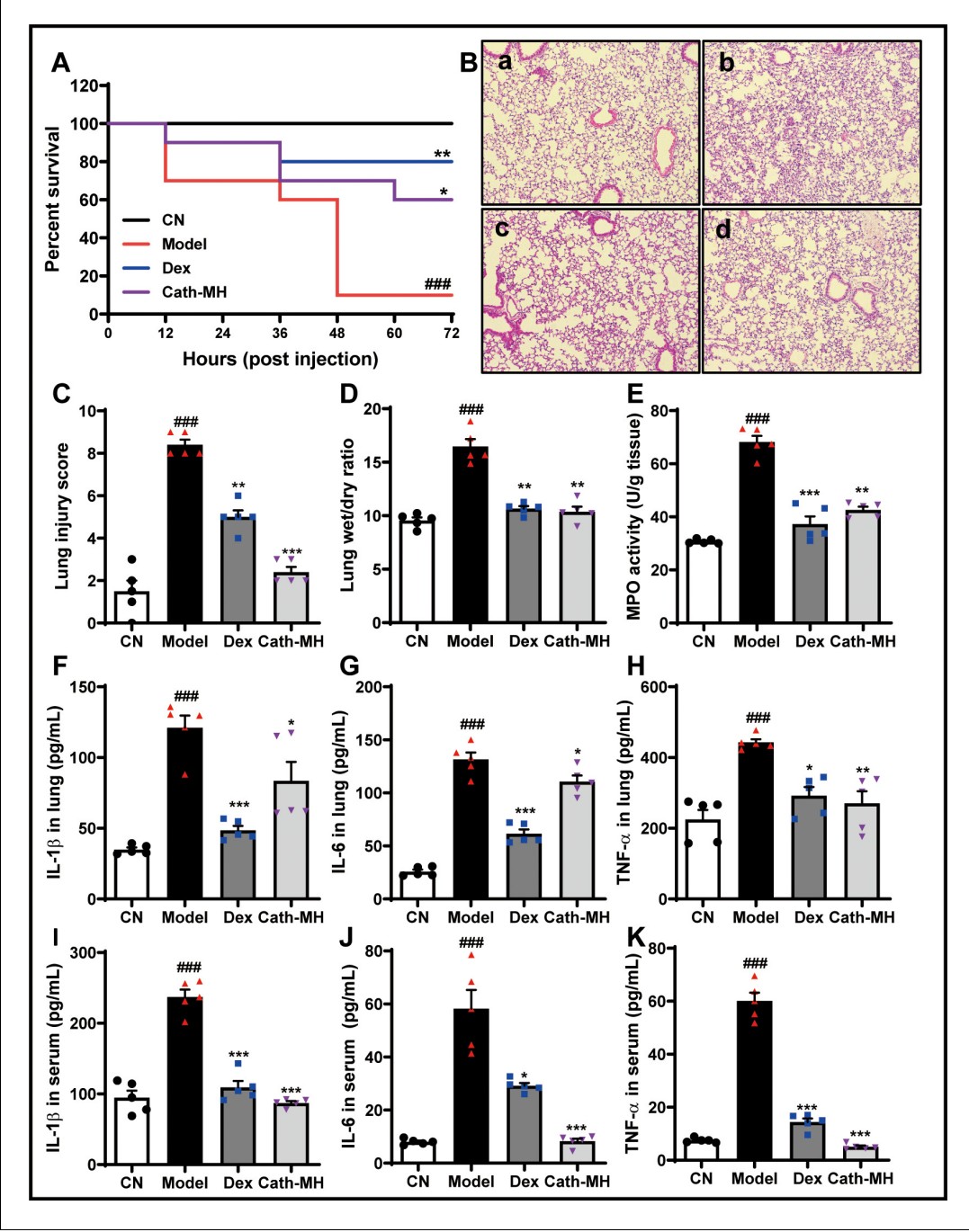

**Figure 6.** Effects of cath-MH on LPS-induced septic mice. Mice except for those in the control group were i.p. challenged with 25 mg/kg LPS 1 hr before injection of saline, cath-MH (5 mg/kg), or dexamethasone (10 mg/kg) (n = 10 mice per group). Survival was observed at the indicated time point after LPS challenge and is expressed as Kaplan–Meier survival curves. Tissues were stained with hematoxylin and eosin, and pro-inflammatory cytokines were measured by ELISA 12 hr after LPS challenge. (**A**) Effects of cath-MH on the survival rates of LPS-induced mice. (**B**) Lung histology after 12 hr of LPS challenge in control (**a**), model (**b**), dexamethasone-treated (**c**), and cath-MH-treated (**d**) mice (magnification ×100). (**C**) Lung injury scores. (**D**) Wet/dry ratio of lung tissues. (**E**) MPO activity of lung tissues. (**F–H**) Effects of cath-MH on pro-inflammatory cytokine protein expression in lung tissues of mice. (**I–K**) Effects of cath-MH on pro-inflammatory cytokine protein expression in serum. Data presented are mean ± SEM (n = 5 per group). CN: control, Dex: dexamethasone. $^{###}p<0.001$ significantly different compared to control group treated with saline only; $^*p<0.05$, $^{**}p<0.01$, $^{***}p<0.001$ compared to the model group.

The online version of this article includes the following figure supplement(s) for figure 6:

*Figure 6 continued on next page*

Notably, cath-MH had a stronger inhibitory effect than dexamethasone on phosphorylated ERK, JNK, and p38 in kidneys, livers, and lungs.

## Effects of cath-MH in CLP-induced septic mice

The mortality rates of cath-MH-treated, CLP model, and sham control mice were 75%, 87.5%, and 0% three days after operation, respectively (*Figure 8A*). On histopathological examination, the lung tissues of mice in the sham control group were intact, and no edema or inflammatory cell infiltrates were observed in alveolar septa, with alveolar cavities clearly visible. Compared with sham control mice, CLP markedly increased alveolar wall thickness and inflammatory cell infiltration accompanied by destruction of the alveolar structures and reductions in the alveolar spaces. In addition, most alveolar cavities of mice in the CLP model group showed exudates, edema, and hemorrhage. However, these changes were alleviated by cath-MH, showing well-preserved alveolar structures and less edema and hemorrhage than untreated controls (*Figure 8B*). In agreement, lung damage scores in the CLP model group and the cath-MH group were significantly higher than those in the sham control group. In the cath-MH-treated group, lung damage scores were significantly lower than those in the CLP model group (*Figure 8C*). The wet to dry lung weight ratio in the CLP model group was significantly higher than in the sham control group. However, this increase was significantly inhibited,

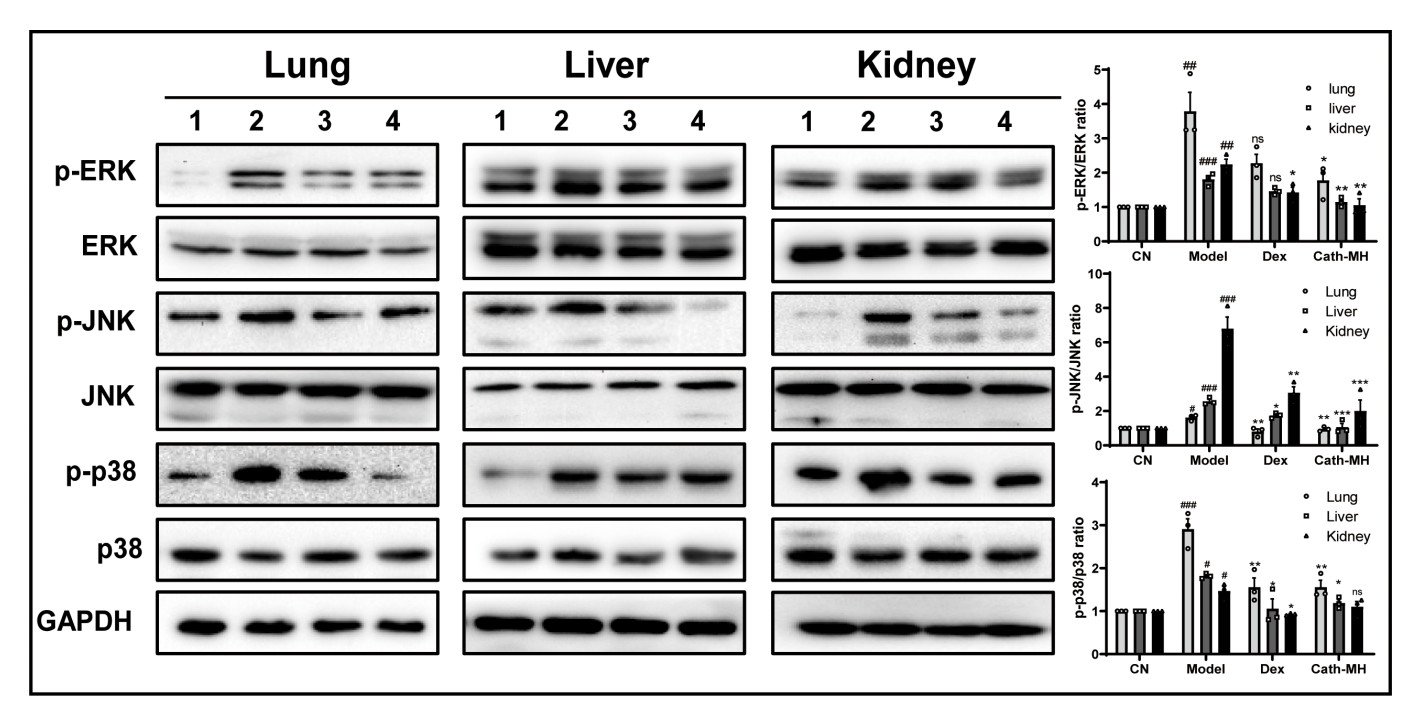

**Figure 7.** Effects of cath-MH on MAPK signaling in lung, liver, and kidney tissues. Mice were euthanized 12 hr after LPS administration, and the tissue samples were prepared for western blotting (n = 3 per group). (A) Western blotting images of p38, ERK, and JNK. (B) Ratio of phosphorylated ERK, JNK, and p38 to total protein. Western blot images shown are representative of three independent experiments with similar results. Images were quantified using ImageJ software. The tissues in Lanes 1–4 are from the control, model, dexamethasone (10 mg/kg), and cath-MH group (5 mg/kg), respectively. Data presented are mean ± SEM (n = 3 per group). #p<0.05, ##p<0.01, ###p<0.001 significantly different compared to the control group; *p<0.05, **p<0.01, ***p<0.001 significantly different compared to the model group.

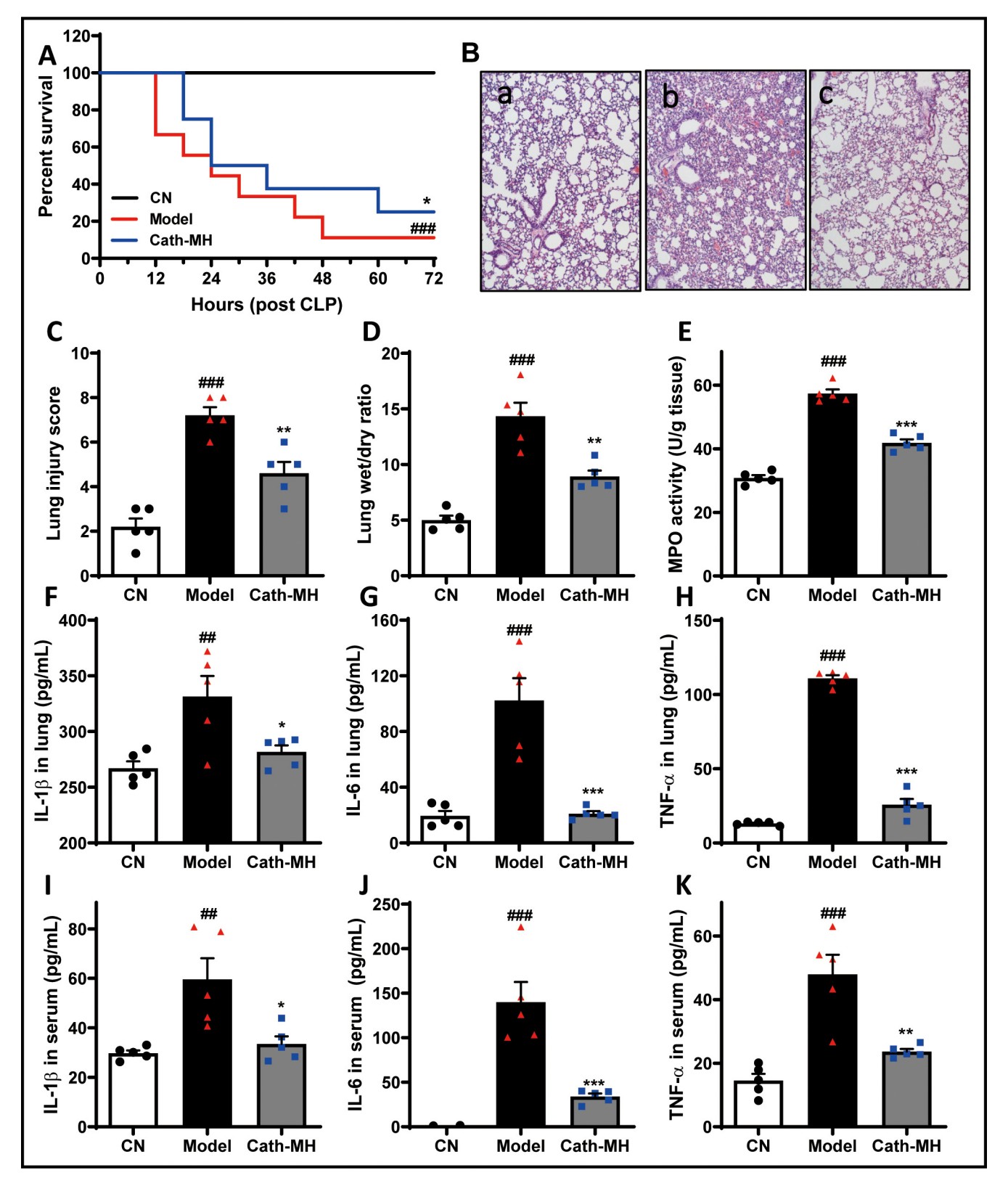

**Figure 8.** Effects of cath-MH in CLP-induced mice. Survival at the indicated time points, pathological alterations, MPO activity, and cytokine expression at 24 hr after CLP were examined. (**A**) Effects of cath-MH on the survival rates of CLP-induced mice. (**B**) Representative histological images of lungs 24
*Figure 8 continued on next page*

**Figure 8 continued**

hr after operation in control (**a**), model (**b**), and cath-MH-treated (**c**) mice (magnification ×100). (**C**) Lung injury scores of mice. (**D**) Wet/dry ratio of lung tissues. (**E**) MPO activity in lung tissues. (**F–H**) Effects of cath-MH on pro-inflammatory cytokine protein expression in lung tissues. (**I–K**) Effects of cath-MH on serum pro-inflammatory cytokine protein expression. Data presented are mean ± SEM (n = 5 per group). ##$p<0.01$, ###$p<0.001$ significantly different compared to sham control group; *$p<0.05$, **$p<0.01$, ***$p<0.001$ compared to the model group.

The online version of this article includes the following figure supplement(s) for figure 8:

**Figure supplement 1.** Effect of cath-MH on bleeding time in CLP-induced septic mice.

**Figure supplement 2.** Toxicity of cath-MH in vivo.

although not completely reversed, by cath-MH (*Figure 8D*). Consistently, MPO activity was significantly increased in the CLP model group compared to the sham control group. However, MPO activity was significantly lower in the cath-MH group compared to model mice (by 57.3 ± 2.6%; *Figure 8E*). Similarly, IL-1β, IL-6, and TNF-α expression was significantly upregulated in CLP mice compared those sham controls, and cath-MH significantly abrogated these increases (*Figure 8F–K*). These observations suggest that cath-MH suppresses inflammatory responses and death caused by CLP-induced polymicrobial infection. It also prolonged bleeding time in mice (*Figure 8—figure supplement 1*).

## The hemolysis and toxicity of cath-MH

The results showed that cath-MH had no hemolysis effect in red blood cells from mice and its hemolysis rate was 2.64 ± 1.40% at a concentration of 212.77 μg/ml. In order to evaluate the acute toxicity of cath-MH to mice, the blood biochemical analyses of normal mice inoculated with phosphate-buffered saline (PBS) or enzyme were conducted, and the histopathological changes of mice tissues were examined. As depicted in *Figure 8—figure supplement 2B–E*, there were no significant differences in serum levels of AST, ALT, BUN, and Cr between the two groups after 24 hr and 48 hr inoculation. Furthermore, no significant histopathological changes were observed in tissues of lung, liver, and kidney between the control and cath-MH-treated mice (*Figure 8—figure supplement 2F*). All mice survived and appeared healthy without any abnormal behavior and obvious differences in body weight gain between the two groups during the 7 days observation period (data not shown). Therefore, the injected cath-MH dose, 10 mg/kg, caused no or little acute toxicity to mice, and there were no significant histopathological changes in tissues between the control and cath-MH-treated mice.

## Discussion

Septic shock is a life-threatening condition caused by systemic inflammatory responses to infection (*Huang et al., 2019*). Multiorgan dysfunction or failure, including acute lung injury, is the most important pathological feature of sepsis. Over the past few decades, despite significant advances in the development of antimicrobial therapies and management strategies, sepsis has become a global health crisis and continues to be one of the main causes of death in intensive care units in developed countries (*Huang et al., 2019*). Thus, new therapies to improve clinical outcomes from sepsis are urgently needed. Here, we identified cath-MH in the skin of *M. heymonsivogt* frogs and demonstrate that it may be a promising new candidate therapeutic for sepsis.

There is increasing evidence that some natural AMPs like cathelicidin with strong antimicrobial and anti-inflammatory activity may be excellent therapeutic candidates for sepsis (*Coorens et al., 2017*; *Giacometti et al., 2004*; *Hu et al., 2016*; *Murakami et al., 2009*; *Scott et al., 2002*; *Song et al., 2015*; *Qin et al., 2019*). NCBI BLAST analysis and multisequence alignment revealed that cath-MH is a cathelicidin peptide. Furthermore, CD and X-ray structural analyses showed that mature cath-MH assumes an primarily amphipathic α-helical structure in membrane-mimicking environments and LPS solutions, which is a common structural characteristic of most cathelicidins from tailless amphibians (*Agier et al., 2015*). Adopting such a helical shape during membrane interactions allows unilateral segregation of its hydrophobic residues and results in membrane penetration, transmembrane pore formation, and bacterial lysis (*Dürr et al., 2006*; *Sancho-Vaello et al., 2017*; *Wei et al., 2013*). In agreement, our antimicrobial mechanistic study showed that cath-MH permeates negatively charged bacterial membranes and disrupts membrane integrity, thereby killing bacteria (*Figure 3*).

As a trigger of inflammation and the dysregulated host response in sepsis (*Salomao et al., 2012*), circulating LPS is known to induce extensive pro-inflammatory cytokine expression including TNF-α, IL-1β, and IL-6, which contribute to the pathophysiology of endotoxic shock such as tissue injury, capillary leakage, circulatory failure, and multiorgan dysfunction (*Van Amersfoort et al., 2003*). Like LL-37, OL-CATH2, Lf-CATH1, cathelicidin-PP, cathelicidin-PY, and cathelicidin-RC (*Dürr et al., 2006*; *Ling et al., 2014*; *Mu et al., 2017*; *Qi et al., 2019*; *Wei et al., 2013*; *Yu et al., 2013*), our compelling evidence demonstrates that cath-MH can neutralize LPS and inactivate bacterial pathogens. Thus, it was rational to speculate that cath-MH, like LL-37, can effectively protect against sepsis-induced death (*Dürr et al., 2006*; *Murakami et al., 2009*; *Scott et al., 2002*). In agreement, cath-MH reduced mortality and tissue damage in LPS-induced septic mice. Concurrently, LPS-induced transcription and production of pro-inflammatory TNF-α, IL-1β, and IL-6 in tissues were also markedly decreased by cath-MH. Intriguingly, cath-MH particularly inhibited lung injury in LPS-induced septic mice, although it also had positive effects in other organs. Previous studies have shown that sepsis-induced lung dysfunction leads to the complete failure of other organs (*Matute-Bello et al., 2008*). In addition, CLP results in polymicrobial infection of the peritoneal cavity that can lead to sepsis, major organ failure, and death (*Rittirsch et al., 2009*), and the inflammatory response in CLP septic mice is similar to that in patients with sepsis. Consistent with the results in LPS-induced sepsis models (*Figure 6*), cath-MH decreased the mortality, lung injury, inflammatory cytokine expression, and inflammatory cell infiltration in CLP septic mice. Overall, these results show that cath-MH is strongly protective against sepsis-induced organ dysfunction such as acute lung injury.

It is generally accepted that TLR signaling contributes to the secretion of pro-inflammatory cytokines in endotoxic shock induced by LPS through activation of downstream MAPK signaling (*Leifer and Medvedev, 2016*; *Salomao et al., 2012*). Consistent with this mechanism, cath-MH significantly suppressed the activation of MAPKs in all tested organs. However, similar to the differential effects of black mulberry leaf or pulp extracts on TNF-α protein and mRNA expression in the liver (*de Pádua Lúcio et al., 2018*), TNF-α protein expression was not increased by LPS or inhibited by cath-MH. Host responses to infection are compartmentalized and cytokines undergo post-transcriptional modifications, which may be responsible for the observed differences in TNF-α protein and mRNA expression in the liver during sepsis (*de Pádua Lúcio et al., 2018*). Nevertheless, our results suggest that the protective effect of cath-MH in experimental sepsis models is associated with LPS binding and the inhibition of pro-inflammatory cytokine production by blocking MAPK signaling.

Although both LL-37 and cath-MH share high secondary structural similarity and can protect mice from lethal endotoxic shock, LL-37 displays low sequence identity with cath-MH and lacks loop2 at the C-terminus as well as a disulfide bridge at the N-terminus (*Figure 1D*). In addition, different to any other known cathelicidins from frogs, more residues in cath-MH are extended in the C-terminus of the disulfide bridge and the KVKQ sequence is in the interval between the conserved K10 and KIK16 residues. Finally, compared to cathelicidin-PY (*Wei et al., 2013*), cath-MH contains a larger helix and five extra hydrogen bonds. Thus, cath-MH might have extra functions to other known cathelicidins. Surprisingly, our results show that cath-MH can inhibit plasma coagulation and tPA, plasmin, thrombin, β-tryptase, and elastase while increasing the activity of chymase, which has not been shown for other cathelicidins. Sepsis is almost invariably associated with coagulation abnormalities that facilitate the recruitment of profibrinolytic enzymes such as uPA, tPA, and plasmin (*Gould et al., 2015*). In addition, a lack of human chymase and its homolog in mouse mast cells can increase FXIIIA expression and activity and decrease bleeding time during sepsis. Therefore, fibrinolytic enzymes and chymase closely correlate with disease severity and fatality in sepsis (*Zeerleder et al., 2007*). Furthermore, neutrophil elastase is associated with pulmonary endothelial injury under LPS-induced endotoxemic conditions (*Suzuki et al., 2019*), and inhibitors such as sivelestat can attenuate sepsis-related acute tissue injury (*Li et al., 2016*). Finally, β-tryptase promotes early pulmonary fibrosis in sepsis-induced lung injury and anti-tryptase treatment has a therapeutic effect in experimental colitis (*Isozaki et al., 2006*; *Villar et al., 2015*). Furthermore, inhibiting coagulation with anticoagulants like antithrombin is beneficial for patients with severe infection and sepsis in experimental and initial clinical trials (*Levi and van der Poll, 2017*). Given the observed anticoagulation effect and protease inhibition/promotion activities of cath-MH, we cannot exclude that these functions also contributed to its anti-sepsis role in vivo. Taken together, all these findings support that cath-MH, with multiple

functions, can effectively protect mice from LPS- and CLP-induced death and injury and is therefore a suitable candidate therapeutic for bacterial sepsis.

Cath-MH forms aggregates and stable dimers in solution and lipid bilayers (*Figure 2—figure supplement 3*). Therefore, unlike other AMPs, it is protected from proteolytic degradation, similar to LL-37, the only cathelicidin peptide in humans, cleaved from a human cationic antimicrobial polypeptide of 18 kDa (hCAP18) (*Dürr et al., 2006*). Nevertheless, synthetic LL-37 is rapidly degraded in periodontitis gingival crevicular fluid, restricting its application as a potential therapeutic in the gingival crevice (*McCrudden et al., 2013*). In purified human lung mast cells, β-tryptase rapidly cleaves LL-37 (*Schiemann et al., 2009*). Since it inhibits or activates proteases, cath-MH is likely to have advantages over LL-37 as an anti-sepsis molecule.

In conclusion, cath-MH from the skin of *M. heymonsivogt* frogs contains single α-helix structure in membrane-mimetic environments and favorable antimicrobial, anti-inflammatory, LPS-binding, and protease-inhibiting activities. In addition, cath-MH markedly decreases the mortality, pathological alterations, and pro-inflammatory cytokine expression in LPS- and CLP-induced septic mice by killing microorganisms, neutralizing LPS, and suppressing MAPK signaling and potentially by interfering with coagulation. Taken together, multi-functional cath-MH may serve as a potent potential therapeutic agent for sepsis.

# Materials and methods

## Animals and ethical statement

Male and female adult *M. heymonsivogt* frogs (n = 3), which are not an endangered or protected species, were captured from the countryside of Guangzhou, Guangdong Province, China (23.12°N, 113.28°E) without the need for specific permissions. After collection, the frogs were humanely euthanized using $CO_2$, and the skin was subsequently sheared and stored in liquid nitrogen until use. Kunming and BALB/c mice (weighing 18–20 g) were bought from the Laboratory Animal Center of Southern Medical University (Guangdong, China) and maintained in plastic cages under standard conditions at 25 ± 2°C and 55 ± 10% humidity, with free access to food and water on a 12 hr light/ dark cycle. The Animal Care and Use Ethics of Southern Medical University (NO. L2019030) authorized all protocols and procedures involving live animals.

## Molecular cloning and characterization of cDNA encoding cath-MH

Total RNA from *M. heymonsivogt* skin was extracted using Trizol reagent (Life Technologies, Carlsbad, CA) according to the manufacturer's protocol. mRNAs were purified from total RNA using oligo (dT) cellulose chromatography (Life Technologies). RNA was quantified with Merinton SMA1000 (Merinton, Ann Arbor, MI). cDNA library construction and PCR amplifications were performed as described previously (*Zeng et al., 2018*). The common nucleotide sequence (5′-WSCRCAGRYC TTCACCTCC-3′) encoding the cathelin domain as a primer coupled to the 5′ PCR primer (5′-AAG-CAGTGGTATCAACGCAGAGT-3′) in the SMART cDNA Library Construction Kit was synthesized to screen the 5′ fragments of cDNAs encoding cath-MH. After the 5′ sequence of cDNAs had been obtained, a sense primer (5′-GGATGATGGGGGGCTGCTTCTGG-3′) was designed and coupled with the 3′ PCR primer CDS III of the SMART cDNA Library Construction Kit to screen the 3′ fragment of cDNAs encoding cath-MH.

## Sequence analysis

The physical and chemical parameters of cath-MH were analyzed through the Bioinformatics Resource Portal (http://www.expasy.org/tools/). The assembled sequences were aligned with ClustalW (http://embnet.vital-it.ch/software/ClustalW.html) on the basis of previously characterized cathelicidins from different amphibian species. The phylogenetic tree was constructed using the neighbor-joining method in Mega 6. The reliability of each branch was checked by 1000 bootstrap replications. The evolutionary distances were calculated with the Poisson correction method and are in the units of the number of amino acid substitutions per site. The numbers on the branches suggest the percentage of 1000 bootstrap samples supporting the branch.

## Peptide synthesis and purification

Cath-MH used for bioactivity tests was synthesized by GL Biochem Ltd. (Shanghai, China). Subsequently, an Inertsil ODS-SP (C18) reverse-phase HPLC column (SHIMAZU, Ōsumi, Japan) was used to purify the crude synthetic peptide with a linear gradient of water and acetonitrile (both containing 0.05% TFA) at a flow rate of 1 ml/min. Once the purity computed on the basis of the ratio of different peak areas was higher than 95%, the peak was pooled, lyophilized, and further verified by MALDI-TOF mass spectrometry (*Figure 2—figure supplement 1*).

## Crystallization and data processing

Freshly synthesized and purified cath-MH was used for crystallization experiments. The optimal peptide concentration for crystallization was obtained using the PCT Pre-Crystallization Test (Hampton Research, Aliso Viejo, CA). The Oryx Nano four crystallization robot (Douglas Instruments, Hungerford, UK) was used for initial screening of crystallization conditions. Crystallization screening was realized using the sitting-drop vapor diffusion technique in 96-well crystallization plates (Swissci MRC 2-drop, Molecular Dimensions Ltd., Newmarket, Suffolk, UK), with application of commercial screens Crystal ScreenHT and PEGRxHT (Hampton Research; Aliso Viejo, CA). Crystallization experiments were performed at 277 K. A peptide in TBS buffer pH 7.5 at the concentration of 2.719 mg/ml was mixed with precipitant solution in ratios of 1:1 and 2:1 and equilibrated against 50 µl precipitant solution in the reservoir. X-ray diffraction data were collected at 100K at the BESSY II electron-storage ring on the MX14.2 beamline operated by the Helmholtz-Zentrum Berlin, Berlin-Adlershof, Germany (*Mueller et al., 2015*) to 1.95 Å resolution. Data were recorded on the Pilatus detector S3 2M (Dectris, Switzerland). Data collected in 3600 images were processed using XDS software (*Kabsch, 2010*) with the XDSAPP graphical user interface (*Sparta et al., 2016*). The search model was generated using the automatic molecular replacement pipeline BALBES (*Long et al., 2008*). The peptide structure was solved by automated model building with BUCCANEER (*Cowtan, 2006*) and by molecular replacement in MOLREP (*Vagin and Teplyakov, 2010*). The structure was refined by REFMAC5 from the CCP4 package (*Murshudov et al., 2011*) and manually modeled using the COOT package (*Emsley et al., 2010*). The MolProbity server (*Williams et al., 2018*) was used for final model geometry validation. Cath-MH assembly was determined by the PDBePISA server, and its structure was visualized using PyMOL (*Delano, 2002*). Complete information about crystallization, data collection, and refinement statistics are provided in *Supplementary file 2* and *3*.

## CD analysis

The secondary structure and stability of cath-MH in solvent environments were determined by CD using a Chirascan plus ACD spectropolarimeter (Applied Photophysics Ltd, Leatherhead, UK). Spectra at 180–260 nm were determined in a 0.1 cm path-length cell with 1 nm bandwidth, 1 s response time, and a scan speed of 100 nm/min at 25°C. Cath-MH at a concentration of 0.213 mg/ml was dissolved in 0, 30, 60, 90, and 120 mM SDS solutions or 60 mM SDS solution plus 0, 100, 200, and 400 mM NaCl, respectively. For temperature stability analysis, 0.213 mg/ml cath-MH dissolved in 60 mM SDS was incubated for 20, 37, 50, 70, and 90°C for 1 hr before CD spectra measurements. CD spectra of 0.213 mg/ml peptides were also measured in the solution containing 50 µM LPS (L2880, *E. coli* $O_{55}:B_5$, Sigma-Aldrich, St. Louis, MO). CD data were expressed as the mean residue ellipticity (θ) of three consecutive scans per sample in deg·cm²/dmol.

## Antimicrobial assay

All ATCC microorganisms were bought from the Guangdong Institute of Microbiology. The antimicrobial capability of cath-MH was evaluated through minimal inhibitory concentrations (MICs) by the twofold microdilution method as reported previously (*Zeng et al., 2018*). In brief, after growing in Muller-Hinton (MH) broth at 37°C to exponential phase, microbes were then diluted with fresh MH broth to reach $10^6$ CFU/ml density. An equal volume of microbial inocula was added into 96-well plates with different concentrations of cath-MH diluted in MH broth. Plates were incubated at 37°C for 14 hr, and the MIC values at which there was no microbial growth were measured at 600 nm with a microplate spectrophotometer (Tecan Trading AG, Männedorf, Switzerland). All experiments were carried out in triplicates with ampicillin and meropenem as positive control.

## Stability analysis

The stability of cath-MH under different conditions was assessed as described in our previous study (*Zeng et al., 2018*). In short, after culturing in MH broth to reach exponential phase, *E. coli* ATCC 25922 was diluted to $10^6$ CFU/ml in fresh MH broth. After pretreatment with 0, 50, 100, 200, and 400 mM NaCl for 1 hr at room temperature, gradient solutions of cath-MH were prepared in MH broth and added to equal volumes of microbial inocula to measure their MICs. In the thermal stability assay, 4.26 mg/ml cath-MH in sterile deionized water was incubated at 20, 37, 50, 70, and 90°C for 1 hr before its MICs against *E. coli* ATCC 25922 were measured. To investigate its serum stability, 8.52 mg/ml cath-MH was incubated in mouse serum at a volume ratio of 1:4 at 37°C for 0–6 hr. The MICs against *E. coli* ATCC 25922 of aliquots at each time point were measured, and all experiments were repeated at least three times.

## Bacterial killing kinetics

Bacterial killing kinetics of cath-MH against *E. coli* ATCC 25922 were detected according to the protocol described in our previous study with minor modifications (*Zeng et al., 2018*). Briefly, *E. coli* were harvested in exponential phase, washed twice with PBS, and diluted to $10^6$ CFU/ml with fresh MH broth. The microorganisms were treated with a serial concentration of peptides (one-, two-, and threefold MIC) and grown at 37°C. Aliquots at different time points (0, 30, 60, 90, 120, 150, and 180 min) were coated on MH agar plates, and the bacterial killing kinetics were determined according to the number of viable colonies after incubation at 37°C for 18 hr. Meropenem and sterile 0.9% NaCl solution were applied as the positive and negative controls, respectively.

## Confocal laser scanning microscopy

Confocal laser scanning microscopy was carried out to determine antibacterial mechanism of cath-MH. *E. coli* ATCC 25922 and *S. aureus* ATCC 25923 were collected in late-logarithmic phase, washed three times with PBS, and subsequently incubated with peptide at a final concentration of 1× MIC at 37°C for 1.5 hr. Confocal laser scanning microscopy was used to observe cells stained with SYTO9 and PI for 30 min at room temperature in the dark. The excitation/emission wavelengths for SYTO9 and PI were 480/500 nm and 490/635 nm, respectively. Green fluorescence represented living microbes stained with SYTO9, while red fluorescence represented dead bacteria stained with PI. About 5–10 single-plane images per coverslip were captured.

## Scanning electron microscopy

Morphological changes in peptide-treated microbes were investigated using scanning electron microscopy using our previously published method (*Zeng et al., 2018*). Briefly, 1× MIC cath-MH was incubated with logarithmic phase culture of *E. coli* ATCC 25922 ($10^6$ CFU/ml) and *S. aureus* ATCC 25923 ($10^7$ CFU/ml) for 30 min at 37°C. Bacteria pellets were collected by centrifuging for 10 min at 1000 rpm and fixed with 4% and 2.5% glutaraldehyde solution at room temperature for 4 hr and at 4°C overnight, respectively. After washing with 0.1 M PBS, bacteria were dehydrated with a series of gradient ethanol concentrations and dehydration using Critical Point Dryers (Quorum, East Sussex, UK) prior to examination. After gold coating, bacterial morphology was captured with a JSM-840 instrument (Hitachi, Tokyo, Japan) at 5 kV with about 5–10 single-plane images per sample.

## Membrane binding assays

Membrane binding assays were undertaken with FITC-labeled cath-MH. In short, logarithmic phase cultures of bacteria (*E. coli* ATCC 25922 and *S. aureus* ATCC 25923) were prepared in PBS at a density of $1 \times 10^6$ CFU/ml and incubated with different concentrations of FITC-labeled cath-MH at 37°C for 30 min. The unbound peptide was washed out with cold PBS containing 1% BSA before incubation. Finally, 200 µl PBS or staining buffer containing more 10,000 cells was drawn for flow cytometry analysis. Cell fluorescence intensity was detected with a FACscan flow cytometer (Becton Dickinson Company, Bedford, MA), representing the ability to bind to cell membranes. Cells without cath-MH treatment were regarded as negative control, and all experiments were repeated at least three times.

## Bacterial agglutination tests

*E. coli* ATCC 25922 was separately cultured to exponential phase in MH broth, washed three times with PBS, and then resuspended in TBS to a final density of $10^9$ CFU/ml. Equal volumes of prepared microbe inocula were incubated with 20 µl cath-MH (onefold MIC) alone or with peptide solution pretreated with different intervention factors (1 mg/ml LPS and 4 mM D-(+)-galacturonic acid) for 30 min under gentle shaking conditions for 1 hr at room temperature. Then, bacteria were dried, fixed on the slides, and stained with Gram's dye, after which agglutination was observed using a microscope (Olympus, 100× objective, Melville, NY).

## ITC assay

The thermodynamics of interactions between peptide binding LPS and D-(+)-galacturonic acid were determined using a Micro Cal PEAQ-ITC (Malvern, UK). Fifty millimolar of PBS pH 7.2 was used to prepare stock solutions of peptides, LPS, and D-(+)-galacturonic acid (48280, Sigma-Aldrich). With constant stirring at 25℃ of 25 injections, 1.5 µl aliquots of peptide at concentration of 4.26 mg/ml were titrated into the sample cell filled with 280 µl of 0.05 mM LPS. In the D-(+)-galacturonic acid binding experiment, 10 mM D-(+)-galacturonic acid in the syringe was titrated into 280 µl of 0.213 mg/ml cath-MH in the cell under the conditions described above. The instrument was operated in high feedback mode. After subtracting the heat caused by dilution, the enthalpy change ($\Delta H$) and equilibrium disassociation constant ($K_d$) were calculated by fitting to a single-site binding model with MicroCal Origin software. The entropy change ($\Delta S$) and Gibb's free-energy change ($\Delta G$) were obtained from the basic thermodynamic equations. The experiment was repeated three times.

## SPRi assay

The PlexArrayTM HT A100 system (Plexera LLC, Bothell, Washington, DC) with bare gold SPRi chip (Nanocapture gold chip, with a gold layer of 47.5 nm thickness) was used to explore the real-time binding reaction of cath-MH with proteases, plasma, LPS, and D-(+)-galacturonic acid as described in our previous paper (*Zeng et al., 2018*). In brief, 8.52 mg/ml cath-MH dissolved in PBS was spotted in multiplex onto the gold chip surface before incubating overnight in a 4℃ humid box according to the manufacturer's instructions. The SPRi chip was mounted on the instrument after being washed with PBS and blocking with 1 M ethanolamine/$H_2O$ solution (pH 8.5) for 30 min. The following program was used to complete the SPRi procedure: stabilizing baseline with PBS; flowing over the chip with the indicated concentrations of proteases, plasma, LPS, or D-(+)-galacturonic acid as the analytes, respectively; washing the chip with PBS, and regenerating the chip with glycine buffer (pH 2.0). Data were analyzed with the PLEXEA data analysis module and ORIGINLab software (OriginLab). The 1:1 Langmuir model of kinetics fitting from three different concentrations was used to calculate the average kinetics of different complexes, and the values from three different spots were used to compute standard deviations. The $K_d$ was obtained on the basis of kinetic constants counted by curve-fitting association and dissociation rates to real-time binding and washing data. The experiment was repeated three times.

## LPS neutralization assay

To explore whether LPS could inhibit the antimicrobial activity of cath-MH against *B. subtilis*, CFU assays were carried out as reported previously with minor modifications (*Zeng et al., 2018*). In short, LPS at graded concentrations of 0–150 µM was mixed with 0.5×, 1×, 2×, and 3× MICs of cath-MH for 30 min at room temperature. The mixtures were separately added into 96-well plates with $10^6$ CFU/ml *B. subtilis* at log-growth phase. One hour after incubation at 37℃, aliquots were sucked out and serially diluted for spreading on MH agar plates. Viable colonies were counted after the plates had been incubated at 37℃ for 16 hr. The LPS without peptide incubation was used as negative control, and all experiments were repeated at least three times.

## Plasma recalcification time determination

The anticoagulant activity of cath-MH was measured by the recalcification time of normal human platelet-free plasma as described in our previous study (*Xu et al., 2016*). In brief, blood was collected from heathy humans with single-use evacuated tubes for blood specimen collection (Jiangsu Yuli Medical Equipment Co., Ltd, China) and centrifuged at 3000 rpm for 15 min. Serial

concentrations of cath-MH (0.4–53 µg/ml) in 50 µl of 0.15 M NaCl were pre-incubated with 50 µl of normal human plasma at 37°C for 5 min, and 50 µl of 25 mM CaCl$_2$ was added to initiate clotting reactions. The clotting time was monitored at 405 nm with a microplate spectrophotometer (Tecan Trading AG, Switzerland). 0.15 M NaCl was used as a negative control. All experiments were performed in triplicate.

## Enzyme inhibition assay

Enzyme inhibition assays were performed as described previously (*Assumpção et al., 2018*). In brief, cath-MH (4.26 µg/ml) was pre-incubated with elastase, chymase, trypsin type III, trypsin type II, thrombin, protease 3, FXIIa, FXIa, FXa, β-tryptase, kallikrein, plasmin, or tPA (tissue plasminogen activator) for 10 min before adding the corresponding substrates at 250 mM final concentration. The hydrolysis rate of the fluorescent substrate was measured with a Tecan Infinite M200 96-well plate fluorescence reader (Tecan Group Ltd) at 365 nm excitation and 450 nm emission wavelengths with a cutoff at 435 nm. The linear fit of the fluorescence increase as a function of time was demonstrated with Magellan Data Analysis Software (Tecan Group Ltd, Männedorf, Switzerland). The observed substrate hydrolysis rate in the presence of cath-MH was compared with that without peptide, which was considered 100% enzyme activity. The different buffers and concentrations of enzymes are shown in *Supplementary file 5*. All assays were performed at 30°C in triplicate.

## Bleeding time

The bleeding time of cath-MH was measured by a tail transection method as described in our previous study with minor modification (*Ma et al., 2011*). In brief, 1 hr after treatment with either tested 5 mg/kg cath-MH or saline water, mice were maintained in a immobilization cage that keeps the tail steady and immersed in 0.9% isotonic saline at 37°C. After 2 min, a distal 2 mm segment of the tail was severed with a razor blade. For the bleeding time of CLP-induced septic mice, the tails were cut down 5 mm at 24 after CLP. The tail was immediately re-immersed in warm saline with the tip of all tails 5 cm below the body. The bleeding time was defined as the time required for the stream of blood to cease. The end point was the arrest of bleeding lasting more than 30 s.

## LPS-induced sepsis in mice

Sepsis was induced by intraperitoneal LPS injection (25 mg/kg) into female BALB/c mice except those in the control group 1 hr before cath-MH (5 mg/kg), dexamethasone (10 mg/kg), or PBS were intraperitoneally administered into septic mice at random. The clinical manifestations of endotoxic shock and mortality were recorded for up to 72 hr after LPS injection. In further experiments, another set of mice were injected and grouped as described above and their lungs, livers, and kidneys surgically isolated 12 hr after LPS injection for macroscopic scoring and ELISA, qRT-PCR, western blotting, and histopathological experiments. The histological injury score criteria were as follows: vascular congestion, edema, and inflammatory cell infiltration for lungs; portal inflammation, hepatocellular necrosis, and inflammatory cell infiltration for livers; and vacuolization and cast formation for kidneys. Each indicator was graded on a scale of 0–3 denoting 'absent', 'mild', 'moderate', and 'severe'. The total injury score was the sum of all indicators (a maximum of 9) (*Kim et al., 2012*).

## MPO determination

MPO activity was determined according to our previous paper with minor modifications (*Kotsyfakis et al., 2006*). Briefly, tissues from septic mice were weighed and homogenized in 50 mM PBS, pH 6.0, containing 0.5% hexadecyltrimethylammonium bromide at 4°C. The samples were freeze-thawed three times followed by sonication and then centrifugation at 12,000 rpm at 4°C for 10 min. Ten microliter supernatants were added into 96-well plates and treated with 90 µl of 50 mM PBS containing 0.167 mg/ml o-dianisidine dihydrochloride and 0.003% hydrogen peroxide. In parallel, dilutions of pure MPO (Sigma-Aldrich) were used to construct a standard curve. The absorbance at 460 nm was determined with a microplate reader. MPO activity in the tissues was shown as units of enzyme/g of tissue. A unit of MPO activity was defined as that catalyzing 1 µM of hydrogen peroxide to water in 1 min at 22°C.

### Histological analysis

Lung, liver, and kidney tissues were fixed with 10% formaldehyde and embedded in paraffin. The samples were then cut into 4 µm thick sections and stained with hematoxylin and eosin and examined with a BX60 microscope (Olympus, Melville, NY) for pathological changes.

### Pro-inflammatory cytokine measurement

After tissue samples from septic mice were homogenized and centrifugated, their supernatants were collected and measurements of nitrite, TNF-α, IL-1β, and IL-6 were performed with commercial kits (Thermo Fisher Scientific Inc, San Diego, CA) according to the manufacturer's instructions.

### Quantitative real-time PCR

After tissue samples from septic mice were homogenized, their total RNAs were extracted with Tri-zol for qRT-PCR to measure the mRNA levels of *IL-6*, *IL-1β*, and *TNF-α* as reported previously by *Zeng et al., 2018*. *GAPDH* expression was quantified as a control to verify equal initial quantities of RNA and as an internal standard to quantify PCR products. Cycle numbers of the target genes were standardized to *GAPDH*, and consequently changes in expression of the target genes were computed. The qRT-PCR primers are shown in *Supplementary file 6*. Forty amplification cycles were required to finish exponential amplification. All experiments were repeated three times.

### Western blot analysis

Tissue samples from septic mice were homogenized in ice-cold PBS with a grinder and then lysed with RIPA lysis buffer (Beyotime Biotechnology, Shanghai, China) at 4℃ for 15 min and centrifuged at 12,000 rpm at 4℃ for 15 min before western blot analysis of the supernatants. Primary antibodies targeting phospho-ERK/ERK, phospho-JNK/JNK, phospho-p38/p38, and GAPDH (1:1500, Cell Signaling Technologies, Danvers, MA) and horseradish peroxidase-conjugated secondary antibodies (1:2000, Cell Signaling Technologies) were applied for western blot analysis. All experiments were repeated three times.

### CLP-induced sepsis in mice

CLP sepsis was induced according to the method described previously (*Rittirsch et al., 2009*). Briefly, six week old BALB/c mice with eight animals in each group were weighed and intraperitoneally anesthetized with 100 mg/kg ketamine plus 8 mg/kg xylazine. The cecum on the left side of the abdomen was isolated under a sterile environment and then was tightly ligated about 1 cm from the end before puncturing two holes in the middle with 22-gauge needles. Subsequently, the cecum was squeezed gently to exteriorize a small amount of feces before returning it to the abdominal cavity. The abdomen was sewn up and then prewarmed 0.9% NaCl (50 ml/kg) was injected subcutaneously to complete resuscitation after the operation. At the same time, a lamp was used to warm the animals for 30 min. cath-MH (5 mg/kg) and PBS were intraperitoneally administered into septic mice at random 1 hr after the operation, respectively. Then, mice were placed in cages with access to food and water for 3 days observation. In further experiments, another set of mice were prepared as described above, and blood and lung tissues were collected 24 hr after surgery for histopathological analysis, dry/wet weight ratios, pathological scoring, and cytokine assays. Sham operation mice were used as negative controls without CLP.

### Acute toxicity in vivo

Briefly, six week old BALB/c mice were weighed and randomly divided into three groups with six animals each, and cath-MH (5 mg/kg and 10 mg/kg) and PBS were intraperitoneally administered into mice of different groups. At 24 hr or 48 hr after injection, serum and organ tissues from mice were collected for blood biochemical and histopathological analysis, respectively.

Data analysis qRT-PCR data were calculated using the $2^{-\Delta\Delta CT}$ method (*Livak and Schmittgen, 2001*). Data were analyzed and plotted with GraphPad Prism 5.0 software (GraphPad Software Inc, La Jolla, CA) as well as Igor and values are presented as mean ± SEM. One-way analysis of variance with post hoc Dunnett test was used to assess significance when comparing two or more groups with a control group and with a post hoc Tukey test when performing comparisons among three or

more groups. The unpaired Student's t-test was used to determine significance between two experimental groups and a p-value≤0.05 was considered statistically significant.

## Additional information

### Funding

| Funder | Grant reference number | Author |
|--------|------------------------|--------|
| National Natural Science Foundation of China | 31772476 | Xueqing Xu |
| National Natural Science Foundation of China | 31861143050 | Xueqing Xu |
| National Natural Science Foundation of China | 31911530077 | Xueqing Xu |
| National Natural Science Foundation of China | 82070038 | Xin Chen |

The funders had no role in study design, data collection and interpretation, or the decision to submit the work for publication.

### Author contributions

Jinwei Chai, Methodology, Writing - original draft, Project administration; Xin Chen, Formal analysis, Writing - review and editing; Tiaofei Ye, Data curation, Writing - review and editing; Baishuang Zeng, Software, Methodology; Qingye Zeng, Data curation, Methodology; Jiena Wu, Data curation; Barbora Kascakova, Software; Larissa Almeida Martins, Tatyana Prudnikova, Methodology; Ivana Kuta Smatanova, Writing - review and editing; Michail Kotsyfakis, Resources, Writing - review and editing; Xueqing Xu, Conceptualization, Supervision, Funding acquisition, Investigation, Project administration

### Author ORCIDs

Larissa Almeida Martins http://orcid.org/0000-0001-8127-6276
Ivana Kuta Smatanova http://orcid.org/0000-0002-1337-9481
Michail Kotsyfakis http://orcid.org/0000-0002-7526-1876
Xueqing Xu https://orcid.org/0000-0002-4525-5803

### Ethics

Animal experimentation: All studies were approved by The Animal Care and Use Ethics of Southern Medical University (NO. L2019030). All studies using animals were conducted following the Animal Research: Reporting In Vivo Experiments (ARRIVE) guidelines and were designed in keeping with the 3Rs (replacement, reduction and refinement) principles.

### Decision letter and Author response

Decision letter https://doi.org/10.7554/eLife.64411.sa1
Author response https://doi.org/10.7554/eLife.64411.sa2

## Additional files

### Supplementary files

- Supplementary file 1. Secondary structural components of cath-MH in different environments.
- Supplementary file 2. Conditions for growing cath-MH crystals used for X-ray diffraction.
- Supplementary file 3. Data collection and refinement statistics of X-ray diffraction.
- Supplementary file 4. Different thermodynamic parameters from ITC.
- Supplementary file 5. The buffers and the enzyme concentrations used in protease inhibition assays.

- Supplementary file 6. Primers (mouse) used for qRT-PCR.
- Transparent reporting form

## Data availability

Sequencing data have been deposited in PDB under accession codes 7AL0. All data generated or analysed during this study are included in the manuscript and supporting files.

The following dataset was generated:

| Author(s) | Year | Dataset title | Dataset URL | Database and Identifier |
|---|---|---|---|---|
| Kascakova B, Prudnikova T, Smatanova IK | 2021 | Crystal Structure of Heymonin, a Novel Frog-derived Peptide | https://www.rcsb.org/structure/7AL0 | RCSB Protein Data Bank, 7AL0 |

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
