## [Decision Letter]

**Acceptance summary:**

In this submission, the authors described the activity of Heymonin both in vitro and in vivo to improve survival from sepsis. Evidence on the potential mechanism of action through interaction with the coagulation system is provided.

**Decision letter after peer review:**

Thank you for submitting your article "Characterization and Functional Analysis of Heymonin, a Novel Frog-derived Peptide with Anti-septicemic Properties" for consideration by *eLife*. Your article has been reviewed by 3 peer reviewers, and the evaluation has been overseen by a Reviewing Editor and Gisela Storz as the Senior Editor. The reviewers have opted to remain anonymous.

The reviewers have discussed the reviews with one another and the Reviewing Editor has drafted this decision to help you prepare a revised submission.

Summary:

In this study, Chai et al describe the identification, structural elucidation, antimicrobial and antiseptic mechanism of a new cathelicidin antimicrobial peptide, heymonin, which was identified from the skin of the frog Microhyla heymonsivogt. The authors show that heymonin suppresses coagulation by affecting the enzymatic activities of tissue plasminogen activator, plasmin, β-tryptase, elastase, thrombin, and chymase, and protects against LPS and CLP-induced sepsis, effectively ameliorating inflammatory response and inflammatory damage in mice through its antimicrobial activity, LPS-neutralizing effect and suppression of MAPK signaling pathway. The study is designed carefully and reports a new cathelicidin with a somewhat unusual function in that it can modulate enzymatic activities of blood components and exhibits a very potent anti-septic shock potential comparable to dexamethasone.

Essential revisions:

• It seems that the binding affinity (Kd) between Heymonin and D-(+)-galacturonic acid is stronger than that with LPS in ITC assay but is converse in SPR assay, the authors should explain the reason.

• The structure data is not closely associated with its function related with sepsis. Primary structural analysis indicated that heynonin possesses 12 basic residues. Secondary structural analysis indicated that heynonin mainly adopts α-helical conformation. Are these structural characterizations responsible for its activities? The current work lacks structure-activity relationship study.

• According to authors description, severe sepsis is associated with abnormal coagulation and it seems that Heymonin can reverse this abnormal process in vivo during the treatment of septic mice. However, the evidence is lack although authors demonstrated that Heymonin can dose-dependently inhibit plasma coagulation in vitro. If they can prove its anti-coagulation effects with tail bleeding experiment in vivo, it will be perfect.

• The author must point out the unique difference between Heymonin and other cathelicidin through data analysis or background search.

• The authors said that the putative cleavage site of the mature heynonin was based on the known frog cathelicidins, and the putative cleavage site of the mature heynonin was between Gly and Ala. But cleavage sites of many known frog cathelicidins were not between Gly and Ala. And the authors should introduce the background of the cleavage sites of known frog cathelicidins in the introduction section.

• The antimicrobial activity of heynonin (Supplementary file 1D) and bacterial killing kinetics of heynonin (Supplementary file 1E) are important results of the study, and I suggest the authors move these results to Figure 4.

• Both circular dichroism analysis of heynonin and crystal structural analysis of heynonin are secondary structural analysis. So I think Figure 2 and Figure 3 need to be incorporated into one figure.

• The authors mentioned that heymonin showed different activities against different enzymes. For example, it increased the enzymatic activity of chymase but significantly inhibited the enzymatic activities of tPA, plasmin, β-tryptase, elastase, and thrombin. As we known, these enzymes belong to serine proteinase. It is rare that a small peptide has such different effect on serine proteases, the authors should explain the reason? In addition, on the basis the fact that heymonin can inhibit plasma coagulation in vitro, the authors think heymonin can inhibit the abnormal coagulation in septic mice which contributes to its antiseptic treatment. Nevertheless, the evidence is lack and its anti-coagulation effects in vivo should be added. Finally, these enzymes are critical serine proteases, which can affect our physiological and pathological states. So the side effects (toxicity) of heynonin need to be evaluated in mice.

• In order to elucidate the binding reaction of heymonin with LPS and D-(+)-galacturonic acid are specific, an isotype peptide (scrambled heymonin) should be introduced in bacterial agglutination tests, ITC assay, and SPRi assay.

---

## [Author Response]

Essential revisions:• It seems that the binding affinity (Kd) between Heymonin and D-(+)-galacturonic acid is stronger than that with LPS in ITC assay but is converse in SPR assay, the authors should explain the reason.

Thanks for your kindly suggestions. ITC and SPR are different technique to examine binding reaction. ITC is a biophysical technique directly characterizing the heat change of binding interactions in solution. However, SPR is a powerful tool monitoring dynamic process of binding and degradation of the reaction between solid and liquid phases (Du X, et al.,2016). Therefore, the interaction between solid and liquid phases is inevitably different with one between two liquid phases. Thus, it is normal to have different binding affinity values measured by two experimental methods, which were also reported by some researchers (Yujia He 2021, Chen, et al.,2014). For example, Yujia He et al. reported that the Kd values of myosin and Hexanal /Octanal detected by SPR were both 5.9 ± 3.9 M ×10^−5^. However, they obtained by ITC were 55.9 ± 15.1 M ×10^−5^ and 26.0 ±1.0 M ×10^−5^, respectively. Thus, we think it reasonable for the discrepancy of dissociation constants from two different assays.

• The structure data is not closely associated with its function related with sepsis. Primary structural analysis indicated that heynonin possesses 12 basic residues. Secondary structural analysis indicated that heynonin mainly adopts α-helical conformation. Are these structural characterizations responsible for its activities? The current work lacks structure-activity relationship study.

We greatly appreciate your valuable and meaningful comments about our manuscript. Sepsis is a fatal disease, characterized by multiple organ failure. Lipopolysaccharide (LPS) plays a crucial role in sepsis and septic shock through hyperactivation of the innate immune system. Heymonin (cath-MH) primarily adopted an amphipathic α-helical structure in membrane-mimicking environments and LPS solutions, which is a common structural characteristic of most cathelicidins from tailless amphibians (Agier et al., 2015). By contrast to defensins, cathelicidin family of AMPs, particularly those who adopt helical structures in lipids, were demonstrated to possess antimicrobial and LPS neutralization activity (Durr et al.,2006; Gennaro et al.,2000; Hirata et al.,1994). Sequence alignment demonstrated 50% sequence similarity between Heymonin (cath-MH) and human LL-37. However, human LL-37 has multifunction. Heymonin (cath-MH) contained more loop than LL-37 in their spatial structure. Generally, the loop structure is assumed to play more role for maintaining extra functions. For example, SMAP-29 contains multiple binding sites and a central hinge which cooperatively to bind LPS with high affinity (B.F. et al.2002).

• According to authors description, severe sepsis is associated with abnormal coagulation and it seems that Heymonin can reverse this abnormal process in vivo during the treatment of septic mice. However, the evidence is lack although authors demonstrated that Heymonin can dose-dependently inhibit plasma coagulation in vitro. If they can prove its anti-coagulation effects with tail bleeding experiment in vivo, it will be perfect.

Thanks for your meaningful suggestion. We have added this experiment in the revised version. As showed in Figure 8—figure supplement 1, the bleeding time of CLP-induced septic mice decreased significantly compared to the control, and Heymonin (cath-MH) treatment group could reverse this change.

• The author must point out the unique difference between Heymonin and other cathelicidin through data analysis or background search.

Thanks for your kindly suggestions. As shown in the following table, Cathelicidins have been shown to kill microbes, neutralize LPS and modulate the activation of several toll-like-receptor (TLR) ligands. Different to any other known cathelicidins from frogs, as shown in Figure 1C, more residues in Heymonin (cath-MH) are extended after the C-terminus of the disulfide bridge and the KVKQ sequence is in the interval between the conserved K10 and KIK16 residues. Furthermore, compared to cathelicidin-PY (Wei et al.,2013) and cathelicidin-RC1 (Ling et al.,2014), Heymonin (cath-MH) contains a larger helix and extra hydrogen bonds. However, compared to LL-37, Heymonin (cath-MH) has an extra loop at its C-terminus (Dürr et al.,2006). The predicted structures of Hc-CATH (Wei et al.,2015) and cathelicidin-RC2 (Ling et al.,2014) exhibit a helix-strand-helix conformation. In addition, cathelicidin-PP mainly adopts a β-sheet structure with a small α-helix (Mu et al.,2017). All of them also are different from Heymonin (cath-MH). Due to its difference in spatial structure with other Cathelicidins, Heymonin (cath-MH) has extra functions to other known cathelicidins such as inhibitory activity of serine proteases.

**Author response table 1. resptable1:** The structure and function of different cathelicidins.

Peptide	Origin	Amino acid sequence	#AA	Function	Ref.
Heymonin	Frog Microhyla heymonsivogt	APCKLGCKIKKVKQKIKQKLKAKVNAVKTVIGKISEHLG	39	antimicrobial activity; LPS neutralization; Sepsis therapeutic efficacy; Serine protease inhibitory activity	None
cathelicidin-PY	frog Paa yunnanensis	RKCNFLCKLKEKLRTVITSHIDKVLRPQG	29	antimicrobial activity; LPS neutralization; anti-inflammatory	(Wei et al., 2013)
human LL-37	human	LLGDFFRKSKEKIGKEFKRIVQRIKDFLRNLVPRTES	37	antimicrobial activity; LPS neutralization; anti- inflammatory; wound healing and angiogenesis; Sepsis therapeutic efficacy; platelet activation induction	(Durr et al., 2006)
Hc-CATH	sea snake Hydrophis cyanocinctus	KFFKRLLKSVRRAVKKFRKKPRLIGLSTLL	30	antimicrobial activity; LPS neutralization; anti-inflammatory	(Wei et al., 2005)
chicken CATH-2	Chicken	RFGRFLRKIRRFRPKVTITIQGSARG	26	antimicrobial activity; immunomodulatory	(Kraaij et al., 2020)
cathelicidin-PP	tree frog Polypedates puerensis	ASENGKCNLLCLVKKKLRAVGNVIKTVVGKIA	32	antimicrobial activity; LPS neutralization; anti-inflammatory	(Mu L et al., 2017)
cathelicidin-RC1	Bullfrog Rana Catesbeiana	KKCKFFCKVKKKIKSIGFQIPIVSIPFK	28	antimicrobial activity	(Ling et al., 2014)
Cathelicidin-RC2	Bullfrog Rana Catesbeiana	KKCGFFCKLKNKLKSTGSRSNIAAGTHGGTFRV	33	weak antimicrobial activity	(Ling et al., 2014)

• The authors said that the putative cleavage site of the mature heynonin was based on the known frog cathelicidins, and the putative cleavage site of the mature heynonin was between Gly and Ala. But cleavage sites of many known frog cathelicidins were not between Gly and Ala. And the authors should introduce the background of the cleavage sites of known frog cathelicidins in the introduction section.

Thanks for your kindly suggestions. The proteolytic maturation is necessary for cathelicidin to manifest its antimicrobial activity (Cole et al., 2001). For mammalian sequences, this cleavage usually occurs through the action of elastase at a valine or alanine or isoleucine. However, other elastase-sensitive sites also exist (Shinnar et al., 2003). For amphibian, most mature bioactive peptides are generated from the precursor by cleavage at a classical -KR- cleavage site located proximal to the N-terminus. But partial peptides are cleaved from their precursors by -RR- or -KK- cleavage sites. Compared with other cathelicidins, most mature peptides are released after the four residues following -KR- cleavage site (Xu and Lai, 2015). In addition, the analysis of physicochemical parameters has showed that the mature peptide of Heymonin (cath-MH) is more positive charged than the sequence with extra more four residues after -KR- cleavage site. With the high cationicity, AMPs can exhibit the initial electrostatic interactions with negatively charged bacterial membrane components (Takahashi et al., 2010). Based on these analysis, we predicted the sequence of mature peptide is APCKLGCKIKKVKQKIKQKLKAKVNAVKTVIGKISEHLG, which is demonstrated right by function assays with synthetized peptide.

• The antimicrobial activity of heynonin (Supplementary file 1D) and bacterial killing kinetics of heynonin (Supplementary file 1E) are important results of the study, and I suggest the authors move these results to Figure 4.

Thanks for your suggestion and we have moved these tables as supplement to Figure 4 and renamed it as Figure 3—figure supplement 1.

• Both circular dichroism analysis of heynonin and crystal structural analysis of heynonin are secondary structural analysis. So I think Figure 2 and Figure 3 need to be incorporated into one figure.

Thanks for your kindly suggestion and we have combined Figure 2 and Figure 3 in the revised manuscript.

• The authors mentioned that heymonin showed different activities against different enzymes. For example, it increased the enzymatic activity of chymase but significantly inhibited the enzymatic activities of tPA, plasmin, β-tryptase, elastase, and thrombin. As we known, these enzymes belong to serine proteinase. It is rare that a small peptide has such different effect on serine proteases, the authors should explain the reason? In addition, on the basis the fact that heymonin can inhibit plasma coagulation in vitro, the authors think heymonin can inhibit the abnormal coagulation in septic mice which contributes to its antiseptic treatment. Nevertheless, the evidence is lack and its anti-coagulation effects in vivo should be added. Finally, these enzymes are critical serine proteases, which can affect our physiological and pathological states. So the side effects (toxicity) of heynonin need to be evaluated in mice.

1. Thanks for your kindly suggestions. Different to any other known cathelicidins from frogs, more residues in Heymonin (cath-MH) are extended in the C-terminus of the disulfide bridge and the KVKQ sequence is in the interval between the conserved K10 and KIK16 residues. In addition, compared to cathelicidin-PY (Wei et al.,2013), Heymonin (cath-MH) contains a larger helix and five extra hydrogen bonds. Thus, Heymonin (cath-MH) might have extra functions to other known amphibian cathelicidins. The Heymonin (cath-MH) functions in vivo as inhibitors of serine protease formation. Human LL-37 is also known to inhibit physiological processes by attacking more than one activating pathway simultaneously (Su et al., 2016) which is associated with a serial of proteases. Different with LL-37, Heymonin (cath-MH) contains an extra loop at C-terminus. Generally, the loop structure is assumed to play more role for maintaining extra functions. The peptide may, therefore, display different activity against enzymes. Of course, we also cannot rule out the possibility that Heymonin (cath-MH) can target to specific location by other way.

The protein docking between Heymonin (cath-MH) and chymase or thrombin were performed by the ZDOCK server and analyzed by PyMOL software, the results showed that Heymonin (cath-MH) can bind both chymase and thrombin by hydrogen bonding (Figure 1A). However, Heymonin (cath-MH) can bind the surface of chymase which is different with other chymase inhibitors, such as sunflower trypsin inhibitor-1 (Li, et al.,2020). However, Heymonin (cath-MH) is in close contact with multiple residues in exosite-I (one of prominent active site cleft of thrombin) as depicted in Figure 1B (Koh, et al.,2011). For the complex of Heymonin (cath-MH) and thrombin, the residues, Lys13, Lys19, Val30 and Ile34 of Heymonin (cath-MH) can form five hydrogen bonds with Asp60, Gln151, Arg75 and Arg78 of the thrombin. In addition, the Lys23 residues of Heymonin (cath-MH) can form four hydrogen bonds (Arg73, Trp141, Glu192 and Asn143) with thrombin residues. Heymonin (cath-MH) bound to the thrombin active site surfaces that interact with substrate residues because the inhibitory site peptide already occupied an external site, which inhibited blood coagulation.

**Author response image 1. sa2fig1:** Model comparison of Heymonin (cath-MH) bound to chymase (A) and thrombin (B). Molecular dynamics simulations were performed to examine the binding interactions of Heymonin (cath-MH) using models generated from the existing X-ray crystal structures of chymase (PDB ID 4AG1) and thrombin (PDB ID 4UD9). The model of Heymonin (cath-MH) (cyans or blue) bound to chymase (A. gray) or thrombin (B. gray) is shown on the left, with the enzyme shown in ribbon representation and Heymonin (cath-MH) shown in stick model (CPK coloring, cyans or blue ) on the right.

2. Thanks for your meaningful suggestion and we have added related assays in the revised manuscript. (Figure 8—figure supplement 1).

3. The hemolysis and toxicity of Heymonin (cath-MH).

The results showed that Heymonin (cath-MH) had no hemolysis effect in red blood cells from mice and its hemolysis rate was 2.64 ± 1.40 at a concentration of 212.77 μg/ml. In order to evaluate the acute toxicity of Heymonin (cath-MH) to mice, the blood biochemical analyses of normal mice inoculated with PBS or Heymonin (cath-MH) were conducted, and the histopathological changes of mice tissues were examined. As depicted in Figure 8—figure supplement 2B-E, there were no significant differences in serum levels of AST, ALT, BUN and Cr between the two groups after 24 hr and 48 hr inoculation. Furthermore, no significantly histopathological changes were observed in tissues of lung, liver and kidney between the control and Heymonin (cath-MH)-treated mice (Figure 8—figure supplement 2F). All mice survived and appeared healthy without any abnormal behavior and obvious differences in body weight gain between the two groups during the 7-days observation period (data not shown). Therefore, the injected Heymonin (cath-MH) dose, 10 mg/kg caused no or little acute toxicity to mice, there were no significantly histopathological changes in tissues between the control and Heymonin (cath-MH)-treated mice.

• In order to elucidate the binding reaction of heymonin with LPS and D-(+)-galacturonic acid are specific, an isotype peptide (scrambled heymonin) should be introduced in bacterial agglutination tests, ITC assay, and SPRi assay.

We greatly appreciate your valuable and meaningful comments about our manuscript. More than one kinds of peptides were spotted onto the gold chip surface in SPRi assay and the results showed that other peptide didn’t produce intense SPRi signal change (data not shown), indicating that these peptides cannot binding to tested proteases. We have tried other peptides to bind LPS and D-(+)-galacturonic acid in ITC assay and they did not bind. In addition, in bacterial agglutination tests, we used BSA as a negative control, too.

References:

1) Du X, Li Y, Xia YL, Ai SM, Liang J, Sang P, Ji XL, Liu SQ. 2016. Insights into protein-ligand interactions: mechanisms, models, and methods. International Journal of Molecular Sciences 17:144, DOI:10.3390/ijms17020144, PMID: 26821017

2) Yujia He, Changyu Zhou, Chunbao Li, Guanghong Zhou. 2021. Effect of incubation temperature on the binding capacity of flavor flavor compounds to myosin. Food Chemistry 346: 128976, DOI: 10.1016/j.foodchem.2020.128976, PMID: 33476948

3) Chen K, Michelsen K, Kurzeja RJ, Han J, Vazir M, St JDJ, Hale C, Wahl RC. 2014. Discovery of small-molecule glucokinase regulatory protein modulators that restore glucokinase activity. Journal of Biomolecular Screening. 19:1014-23, DOI: 10.1177/1087057114530468, PMID: 24717911

4) Agier J, Efenberger M, Brzezinska-Blaszczyk E. 2015. Cathelicidin impact on inflammatory cells. Central European Journal of Immunology 40:225-35, DOI: 10.5114/ceji.2015.51359, PMID: 26557038

5) Durr, U.H.; Sudheendra, U.S.; Ramamoorthy, A. 2006. LL-37, the only human member of the cathelicidin family of antimicrobial peptides. Biochim Biophys Acta 1758: 1408-1425. DOI: 10.1016/j.bbamem.2006.03.030, PMID: 16716248

6) Gennaro, R.; Zanetti, M. 2000. Structural features and biological activities of the cathelicidin-derived antimicrobial peptides. Biopolymers 55:31-49. DOI: 10.1002/1097-0282(2000)55:1<31::AID-BIP40>3.0.CO;2-9, PMID: 10931440

7) Hirata, M.; Shimomura, Y.; Yoshida, M.; Wright, S.C.; Larrick, J.W. 1994. Endotoxin-binding synthetic peptides with endotoxin- neutralizing, antibacterial and anticoagulant activities. Prog. Clin. Biol. Res. 388, 147-159. PMID: 7831355

8) B.F. Tack, M.V. Sawai, W.R. Kearney, A.D. Robertson, M.A. Sherman, W. Wang, T. Hong, L.M. Boo, H. Wu, A.J. Waring, R.I. Lehrer 2002. SMAP29 has two LPS-binding sites and a central hinge. European Journal of Biochemistry 269:1181-1189. DOI: 10.1046/j.0014-2956.2002.02751.x, PMID: 11856344

9) Mu L, Zhou L, Yang J, Zhuang L, Tang J, Liu T, Wu J, Yang H. 2017. The first identified cathelicidin from tree frogs possesses anti-inflammatory and partial LPS neutralization activities. AMINO ACIDS 49:1571-1585. DOI: 10.1007/s00726-017-2449-7, PMID: 28593346

10) Wei L, Yang J, He X, Mo G, Hong J, Yan X, Lin D, Lai R. 2013. Structure and function of a potent lipopolysaccharide-binding antimicrobial and anti-inflammatory peptide. Journal of Medical Biochemistry 56:3546-3556. DOI: 10.1021/jm4004158, PMID: 23594231

11) Durr, U.H.; Sudheendra, U.S.; Ramamoorthy, A. 2006. LL-37, the only human member of the cathelicidin family of antimicrobial peptides. Biochim Biophys Acta 1758: 1408-1425. DOI: 10.1016/j.bbamem.2006.03.030, PMID: 16716248

12) Wei L, Gao J, Zhang S, Wu S, Xie Z, Ling G, Kuang YQ, Yang Y, Yu H, Wang Y. 2015. Identification and Characterization of the First Cathelicidin from Sea Snakes with Potent Antimicrobial and Anti-inflammatory Activity and Special Mechanism. Journal of Biological Chemistry 290:16633-16652. DOI: 10.1074/jbc.M115.642645, PMID: 26013823

13) Kraaij MD, van Dijk A, Scheenstra MR, van Harten RM, Haagsman HP, Veldhuizen E. 2020. Chicken CATH-2 Increases Antigen Presentation Markers on Chicken Monocytes and Macrophages. Protein and Peptide Letters 27:60-66. DOI: 10.2174/0929866526666190730125525, PMID: 31362652

14) Ling G, Gao J, Zhang S, Xie Z, Wei L, Yu H, Wang Y. 2014. Cathelicidins from the bullfrog Rana catesbeiana provides novel template for peptide antibiotic design. PLOS ONE 9:e93216. DOI: 10.1371/journal.pone.0093216, PMID: 24675879

15) Cole AM, Shi J, Ceccarelli A, Kim YH, Park A, Ganz T. 2001. Inhibition of neutrophil elastase prevents cathelicidin activation and impairs clearance of bacteria from wounds. Blood 97:297-304. DOI:10.1182/blood.v97.1.297, PMID: 11133774

16) Shinnar, A.E., Butler, K.L., Park, H.J., 2003. Cathelicidin family of antimicrobial peptides: proteolytic processing and protease resistance. Bioorganic Chemistry 31: 425e436. DOI:10.1016/S0045-2068(03)00080-4, PMID: 14613764

17) Xu X, Lai R. 2015. The chemistry and biological activities of peptides from amphibian skin secretions. Chemical Reviews 115:1760-1846. DOI:10.1021/cr4006704, PMID: 25594509

18) Takahashi D, Shukla SK, Prakash O, Zhang G. 2010. Structural determinants of host defense peptides for antimicrobial activity and target cell selectivity. BIOCHIMIE 92:1236-1241. DOI:10.1016/j.biochi.2010.02.023, PMID: 20188791

19) Wei L, Yang J, He X, Mo G, Hong J, Yan X, Lin D, Lai R. 2013. Structure and function of a potent lipopolysaccharide-binding antimicrobial and anti-inflammatory peptide. Journal of Medical Biochemistry 56:3546-3556. DOI: 10.1021/jm4004158, PMID: 23594231

20) Su W, Chen Y, Wang C, Ding X, Rwibasira G, Kong Y. 2016. Human cathelicidin LL-37 inhibits platelet aggregation and thrombosis via Src/PI3K/Akt signaling. Biochemical and Biophysical Research Communications 473:283-289. DOI: 10.1016/j.bbrc.2016.03.095, PMID: 27012197

21) Li CY, Yap K, Swedberg JE, Craik DJ, de Veer SJ. 2020. Binding Loop Substitutions in the Cyclic Peptide SFTI-1 Generate Potent and Selective Chymase Inhibitors. Journal of Medical Biochemistry 63:816-826. DOI: 10.1021/acs.jmedchem.9b01811, PMID: 31855419

22) Koh CY, Kumar S, Kazimirova M, Nuttall PA, Radhakrishnan UP, Kim S, Jagadeeswaran P, Imamura T, Mizuguchi J, Iwanaga S, Swaminathan K, Kini RM. 2011. Crystal structure of thrombin in complex with S-variegin: insights of a novel mechanism of inhibition and design of tunable thrombin inhibitors. PLOS ONE 6:e26367. DOI:10.1371/journal.pone.0026367, PMID: 22053189